# Cas9-mediated knockout of Ndrg2 enhances the regenerative potential of dendritic cells for wound healing

Dominic Henn[1,2,3], Dehua Zhao[4,10], Dharshan Sivaraj [1,3,10], Artem Trotsyuk[1,3], Clark Andrew Bonham[1], Katharina S. Fischer[1,3], Tim Kehl [5], Tobias Fehlmann [6], Autumn H. Greco [1], Hudson C. Kussie[1,7], Sylvia E. Moortgat Illouz[1], Jagannath Padmanabhan [1], Janos A. Barrera [1], Ulrich Kneser[8], Hans-Peter Lenhof[5], Michael Januszyk[1], Benjamin Levi[7], Andreas Keller [5], Michael T. Longaker[1], Kellen Chen [1,3,11], Lei S. Qi [4,9,11] ✉ & Geoffrey C. Gurtner [1,3,11] ✉

Chronic wounds impose a significant healthcare burden to a broad patient population. Cell-based therapies, while having shown benefits for the treatment of chronic wounds, have not yet achieved widespread adoption into clinical practice. We developed a CRISPR/Cas9 approach to precisely edit murine dendritic cells to enhance their therapeutic potential for healing chronic wounds. Using single-cell RNA sequencing of tolerogenic dendritic cells, we identified N-myc downregulated gene 2 (*Ndrg2*), which marks a specific population of dendritic cell progenitors, as a promising target for CRISPR knockout. *Ndrg2*-knockout alters the transcriptomic profile of dendritic cells and preserves an immature cell state with a strong pro-angiogenic and regenerative capacity. We then incorporated our CRISPR-based cell engineering within a therapeutic hydrogel for in vivo cell delivery and developed an effective translational approach for dendritic cell-based immunotherapy that accelerated healing of full-thickness wounds in both non-diabetic and diabetic mouse models. These findings could open the door to future clinical trials using safe gene editing in dendritic cells for treating various types of chronic wounds.

More than 6.5 million patients suffer from chronic wounds in the United States, which impact nearly 15% of Medicare beneficiaries at an annual cost estimated at $28–32 billion[1,2]. Chronic wounds constitute a significant burden to the healthcare system as a whole and on the individual level, causing a profoundly negative psychosocial impact on affected patients. Treatment approaches such as skin grafting or tissue engineered skin substitutes are helpful in clinical practice, but often require surgical intervention and frequently fail in the setting of

[1]Hagey Laboratory for Pediatric Regenerative Medicine, Division of Plastic and Reconstructive Surgery, Stanford University, Stanford, CA, USA. [2]Department of Plastic Surgery, University of Texas Southwestern Medical Center, Dallas, TX, USA. [3]Department of Surgery, University of Arizona, Tucson, AZ, USA. [4]Department of Bioengineering, Sarafan ChEM-H, Stanford University, Stanford, CA, USA. [5]Center for Bioinformatics, Saarland Informatics Campus, Saarland University, Saarbrücken, Germany. [6]Chair for Clinical Bioinformatics, Saarland University, Saarbruecken, Germany. [7]Department of Burn, Trauma, Acute and Critical Care Surgery, University of Texas Southwestern Medical Center, Dallas, TX, USA. [8]Department of Hand, Plastic, and Reconstructive Surgery, BG Trauma Center Ludwigshafen, Ruprecht-Karls-University of Heidelberg, Heidelberg, Germany. [9]Chan Zuckerberg Biohub - San Francisco, San Francisco, CA, USA. [10]These authors contributed equally: Dehua Zhao, Dharshan Sivaraj. [11]These authors jointly supervised this work: Kellen Chen, Lei S. Qi, Geoffrey C. Gurtner. ✉e-mail: slqi@stanford.edu; ggurtner@stanford.edu

complex wounds in patients with diabetes and peripheral vascular disease (PVD)[3]. These conditions produce a dysfunctional healing response, characterized by an impaired ability to regenerate a functional microvasculature through the process of angiogenesis[4].

Cell-based therapies have been under development for decades and have demonstrated secretory, immunomodulatory, and regenerative effects on chronic wounds[5]. However, unlike in oncology, treatment of chronic wounds with autologous cells has not been successfully translated into clinical practice, let alone become part of the standard of care for complex wounds. The "workhorses" of current cell therapy approaches in regenerative medicine are mesenchymal stromal cells (MSCs) which, despite their beneficial effect on wound healing, have several disadvantages. Isolation of MSCs requires tissue biopsies or surgical interventions such as liposuction[6], which are not indicated in the vast majority of patients presenting to wound clinics; moreover, stem cell cultivation and expansion are costly[5,7].

Dendritic cells (DCs) are hematopoietic cells that play critical roles in regulating both innate and adaptive immune responses. Depending on their activation status, DCs can promote peripheral immune tolerance (tolerogenic DCs), thus limiting the activation of the immune system and tissue damage[8,9]. For clinical applications, DCs can be easily isolated from the peripheral blood in large quantities using established good manufacturing practice (GMP) protocols for leukaphereses[10]. Multiple clinical trials have reported beneficial effects of tolerogenic DC therapy in the treatment of autoimmune diseases, such as multiple sclerosis, rheumatoid arthritis, type 1 diabetes, Crohn's disease, as well as transplant rejection[11-17]. In addition, Sipuleucil-T, a DC vaccine, demonstrated improved survival in patients with castration-resistant prostate cancer[18], leading to its approval by the US Food and Drug Administration (FDA) in 2010[18,19]. Despite these major clinical advances in DC-based immunotherapy, the application of DCs for wound healing has not been investigated.

CRISPR/Cas9 gene editing has revolutionized the development of T-cell therapeutics for enhanced anti-cancer immunity in hematologic malignancies and solid tumors, demonstrating improved survival in multiple clinical trials[20-23]. In addition, gene editing approaches have been shown to improve the efficacy of MSCs for wound healing[24]. Whether CRISPR/Cas9 technology can modulate and enhance the therapeutic effect of DC immunotherapy has not been investigated so far. Here, we present a CRISPR/Cas9 based approach to enhance the therapeutic potential of DCs in order to develop an effective and translatable cell-based therapy for chronic wounds. Using single-cell RNA sequencing (scRNA-seq) of tolerogenic DCs, we identified N-myc downregulated gene 2 (*Ndrg2*), which marks a specific population of DC progenitors, as a promising target for CRISPR knockout (KO). KO of *Ndrg2* alters the transcriptomic profile of DCs, preserving an immature cell state with a strong pro-angiogenic and regenerative capacity. We further combined our in vitro engineered cells with a therapeutic hydrogel for in vivo cell delivery and developed an effective translational approach for DC therapy of chronic wounds.

## Results

### Ndrg2 expression is reduced in tolerogenic DCs
To identify potential targets for gene editing in DCs, we compared transcriptomic profiles of treatment induced tolerogenic DCs, which confer a variety of clinical benefits[11-17] with untreated DCs. Bone marrow-derived DCs were cultivated from wild-type (WT) mice (C57/BL6) according to standard protocols[25]. Vitamin D3 (calcitriol, VD3) was used to induce tolerogenic DCs, and comparative scRNA-seq of VD3 stimulated and untreated DCs was performed (Fig. 1a, b). DC identity of the cells was confirmed by an expression of the characteristic DC markers *Itgax* (encoding CD11c) and *H2-Ab1* (murine MHC-II gene) (Supplementary Fig. 1a, b). Seven transcriptionally distinct clusters were identified (Fig. 1b and Table S1). Among these, C1, C3, and C4 resembled premature BM-DCs and were characterized by high

expression of *Cd34*, *Csf1r* (encoding CD115), *Ccr2*, and *Cx3cr1*[26]. C0 showed a mature BM-DC expression profile characterized by *Flt3* (encoding CD135)[27] and *Ccr7*[28,29] (Fig. 1b, Supplementary Fig 1c). Other clusters included a proliferating cell cluster (C2) with high expression of the proliferation marker *Stmn1* (stathmin-1)[30], C5 characterized by expression of *Tcirg1* which inhibits T-cell activation[31], and cluster 6 which expressed *Ndrg2*. The smallest cluster (C7) was a mixed cell population with expression of *Prss34* and *Mcpt8*, likely representing basophilic granulocytes that are often found as a by-product in bone marrow cultures (Fig. 1b and Supplementary Fig 1d)[32].

We found that tolerogenic DCs induced by VD3 showed a downregulation of *Ndrg2*, which has previously been reported to inhibit growth factor expression, angiogenesis, and cell proliferation[33-37]. In accordance, tolerogenic DCs showed an overexpression of vascular endothelial growth factor A (*Vegfa*), indicating pro-angiogenic properties potentially associated with *Ndrg2* inhibition. Thus, we identified *Ndrg2* as an interesting target for further investigation (Fig. 1c).

### Ndrg2 marks DC progenitors
Among the untreated DCs, the strongest *Ndrg2* expression was found in cluster 6 which co-expressed the hematopoietic stem cell marker *Cd34*, and *Csf1r* (encoding CD115), as well as *Flt3* (encoding CD135) and *Clec9a* (encoding DNGR-1). This expression profile is consistent with previously defined myeloid progenitor populations, namely MDP (macrophage and dendritic cell progenitor) and CDP (common dendritic cell progenitor), for which phenotypic overlap has been described[38-40]. Interestingly, Ndrg2+ progenitors also expressed *Itage* (CD103) while being negative for *Cd8* (Fig. 1d, e; Supplementary Figs 1d, 2b).

Compared to untreated cells, VD3 treatment almost completely abrogated the expression of *Ndrg2* and its co-expressed genes *Cd34*, *Csd1r*, *Flt3*, *Itgae*, and *Clec9a*, while inducing *Kit* expression (Fig. 1e, Supplementary Fig 2a, b). This was further confirmed with flow cytometry showing 3% of all CD34+ DC progenitors in untreated cultures co-expressed Flt3 (CD135) and CD103, while this population was reduced to 0.8% after VD3 treatment (Fig. 1f).

### Ndrg2 inhibition in DCs promotes EC tube formation in co-cultures
To investigate whether inhibition of *Ndrg2* promotes pro-angiogenic behavior in DCs, we co-cultured BM-DCs with endothelial cells (ECs) and analyzed their angiogenic capacity in EC tube formation assays. *Ndrg2* was inhibited in DC monocultures according to established protocols using a combination of VD3 and lipopolysaccharide (LPS) or dexamethasone (Fig. 1g)[33]. After 24 h of treatment, DCs were co-cultured with ECs in tube formation assays. We found that Ndrg2-inhibited DCs stimulated EC tube formation in vitro, as indicated by a higher EC branch and junction number, longer branch length, and higher total mesh area compared to EC co-cultures with untreated DCs and ECs cultured in basal media (Fig. 1i and Supplementary Fig. 2c, d). The strongest angiogenic response was found in co-cultures with VD3-treated DCs (Fig. 1i, h). This was confirmed in a Luminex multiplex ELISA for 43 proteins performed on culture media of DC monocultures, which showed that DCs treated with VD3 had the highest VEGFA secretion compared to untreated DCs, DCs treated with LPS, or even DCs treated with a combination of VD3 and LPS (Fig. 1j).

Given the strong expression of *Ndrg2* in DC progenitors and the pro-angiogenic response created by pharmacologic *Ndrg2* inhibition, we hypothesized that KO of *Ndrg2* would alter the transcriptional fate and development of BM-DCs to generate cells that would improve the wound healing process. Furthermore, targeted KO of *Ndrg2* might avoid pleiotropic effects of VD3, in which unpredictable off-target or systemic effects could occur, and would also precisely interrogate whether the observed pro-angiogenic features of tolerogenic DCs were in fact due to loss of *Ndrg2* expression.

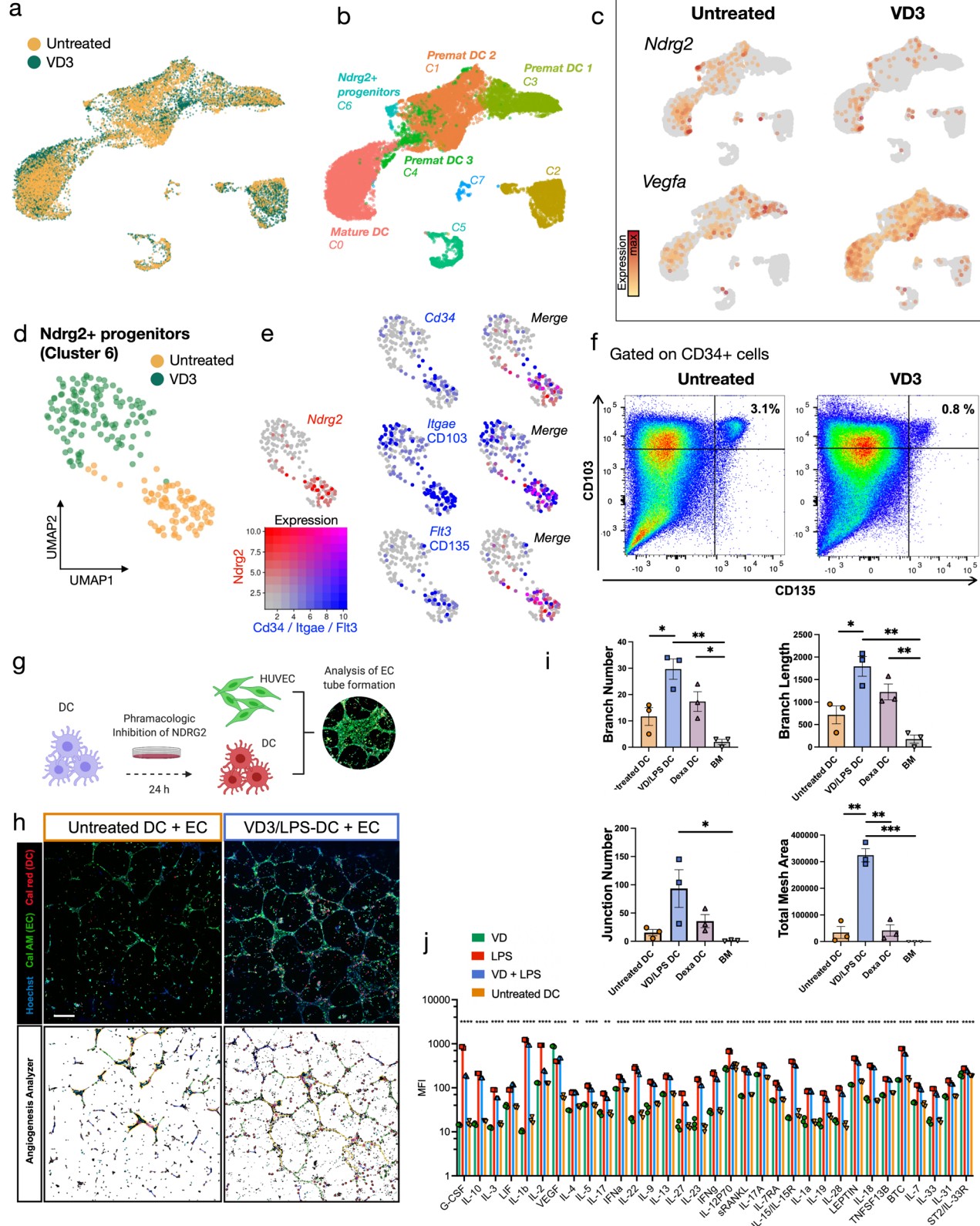

## Development of a CRISPR/Cas9 platform for KO of Ndrg2 in primary DCs

To develop an approach for CRISPR/Cas9 KO of *Ndrg2* in primary murine BM-DCs, we used nucleofection with ribonucleoprotein (Cas-RNP) complexes assembled from Cas9 protein and single guide RNA (sgRNA) (Fig. 2a, b). This technique for CRISPR/Cas9 gene editing has recently been demonstrated to be very effective for editing primary murine and human DCs[41,42], and has several advantages compared to previously employed lentiviral vector-based approaches[43]. To achieve the highest editing efficiency, we compared indels and KO rates after Cas9-RNP nucleofection of DCs using two sets of three sgRNAs from different manufacturers targeting the *Ndrg2* genomic locus, as well as different numbers of cells per reaction and different RNP concentrations (Supplementary Fig. 3a, Table S2). After optimization, using

**Fig. 1 | Ndrg2 inhibition in dendritic cells using vitamin D3 promotes endo-thelial tube formation in co-cultures. a** Single-cell RNA sequencing (scRNA-seq) of vitamin D3 (VD3) stimulated and untreated bone marrow-derived dendritic cells (BM-DCs) (10X Genomics Chromium), cells colored by experimental group. **b** Cells colored by Seurat clusters. Premat = premature. **c** *Vegfa* and *Ndrg2* expression projected onto UMAP embedding in untreated and VD3 stimulated cells. **d** Subset of cluster 6 showing Ndrg2+ progenitor cells, colored by experimental group. **e** Expression of *Ndrg2* (red) and *Cd34*, *Itgae*, and *Flt3* (blue) projected onto UMAP embedding of cluster 6 subset. Right column shows merged expression of Ndrg2 and co-expressed genes. **f** Flow cytometry of untreated and VD3 stimulated BM-DCs, gated on CD34, CD103, and CD135. **g** Schematic of co-culture experiments using BM-DCs and human umbilical vein endothelial cells (HUVECs). **h** Endothelial tube formation after co-culture of endothelial cells together with untreated DCs or

VD3 and lipopolysaccharide (LPS) stimulated DCs. DCs were stained with calcein red, ECs were stained with calcein AM (green). Cell nuclei were stained with Hoechst (blue). Scale bar: 200 μm. Bottom row shows binary images created using the Angiogenesis Analyzer. **i**, Analysis of HUVEC branch number, branch length (pixels), junction number and total mesh area (pixels), (*n* = 3 biological replicates, one-way analysis (ANOVA) with Tukey's multiple comparisons test: *$P < 0.05$, **$P < 0.01$, ***$P < 0.001$. Data are presented as mean values ± SEM.). **j** Luminex multiplex ELISA showing protein expression of 33 differentially expressed proteins in cell culture media of untreated DCs and of DCs stimulated with either VD3, LPS or VD3 and LPS (one-way analysis (ANOVA) with Tukey's multiple comparisons test: **$P < 0.01$, ****$P < 0.0001$, *n* = 3 biological replicates. Data are presented as mean values ± SEM). Source data are provided as a Source Data file.

GFP-fused Cas9 protein (dCas9-GFP) and flow cytometry, we confirmed a >90% transfection efficiency of DCs with our approach (Fig. 2c). Sanger sequencing and the Interference of CRISPR Editing (ICE) tool (Synthego) were used to compute indel and KO rates, which were 91% resp. 88% using a triple-guide RNP approach (Fig. 2d, Supplementary Fig. 3a, "Methods"). We also performed immunocytochemical staining of CRISPR-edited (Ndrg2-KO) and control (untreated) DCs in vitro and observed a significant reduction in Ndrg2 protein expression (Fig. 2e). Single-cell RNA-seq of Ndrg2-KO and control DCs showed that the edited cells retained their DC identify and express common DC markers such as *Ptprc* (encoding CD45), *Itgax* (encoding CD11c), and *H2-Ab1* (encoding MHCII) (Supplementary Fig. 3b)[44].

## Gene editing with Cas9-RNP causes no observable off-target effects

To analyze specific mutations introduced by CRISPR gene editing in DCs at the target site as well as potential off-target effects, we performed whole genome sequencing (WGS) with an ultrahigh depth (~100X coverage) of CRISPR-edited DCs using our triple-guide RNP approach (Fig. 2f). Wild-type DCs were analyzed as controls. A variety of different on-target effects were identified, the most common of which were upstream gene variants, followed by 5' UTR variants (Fig. 2g). The Cas-OFFinder algorithm which allows for unbiased extensive mutation searching, was used to identify potential off-target sites. Importantly, indels were not detected within 15 base pairs (bp) up- and downstream of the sites predicted by Cas-OFFinder[45]. When each site was broadened to 200 bp up- and downstream, indels in 14 sites were identified (Fig. 2h, Table S3). By manual inspection of these loci, we found that only 7 loci showed indels that occurred only in the Ndrg2-KO group and not in the control group. After identifying the most likely off-target sites, we further aligned the sequence between the predicted Cas-OFFinder cut sites and the three sgRNA sequences targeting *Ndrg2*. Among the three sites that showed the highest alignment, we only identified a protospacer adjacent motive (PAM) in two of the sites. All other predicted off-target sites did not show similarity to the sgRNA sequences and therefore are unlikely true off-target sites (Supplementary Fig. 3c, d). This set of analyses confirmed that the designed guide RNAs and the use of RNP in DCs cells produced negligible to no observable off-target effects.

## Ndrg2-KO prevents DC maturation and induces regenerative gene expression profiles

To assess the impact of Ndrg2-KO on the transcriptional fate of DCs, we performed scRNA-seq of CRISPR-edited DCs and compared their transcriptomic profiles with those of untreated DCs and DCs treated with VD3 (Fig. 1a). A total of 23,871 cells were analyzed (Fig. 3a). We employed RNA velocity analysis using dynamical modeling with the scVelo package to determine changes in DC differentiation in response to Ndrg2-KO[46]. A differentiation stream was identified originating from the Ndrg2+ progenitors (cluster 9) that progressed through clusters 1 and 3 and ended in cluster 0 (Fig. 3a, b and

Table S4). Most Ndrg2-KO cells were found in cluster 1, which was characterized as a premature DC cluster with strong expression of *Csf1r*, *Ccr2*, and *Cx3cr1* (Supplementary Fig. 5a)[26], whereas control and VD3-treated cells mostly localized to cluster 0, which was identified as a mature cluster with strong expression of *Ccr7, Flt3, Cd80, and Cd83* (Fig. 3c, Supplementary Fig. 5c)[27–29]. Cluster 3 was identified as a smaller mature cluster which mostly contained Ndrg2-KO cells and strongly expressed *Nr4a3* (Supplementary Fig. 5d). Clusters 2, 7, and 8 were identified as premature clusters that mostly contained control and VD3-treated cells and expressed *Cx3cr1* and *Csf1r* (cluster 7) as well as *S100a9* (clusters 2 and 8) and *Stamn1* (cluster 8) (Supplementary Fig. 5a, e). Cluster 4 contained the *Stmn1* expressing proliferating cells mostly from the VD treated group, that had been identified in Fig. 1b (Supplementary Fig. 5e). Cluster 11 represented a very small cluster of mixed cells as a by-product of the bone marrow cultures.

Using CytoTRACE, we compared the relative differentiation states of individual cells based on the distribution of unique mRNA transcripts and found that cluster 1, which mostly contained Ndrg2-KO cells, was among the least differentiated clusters, whereas cluster 0, which contained mostly VD3-treated and untreated cells, showed more advanced differentiation states (Fig. 3d)[47]. In accordance, KO of *Ndrg2* led to a downregulation of the DC maturation markers *Cd80* and *Cd83* and an upregulation of *Csf1r*, which marks premature DCs (Fig. 3e). Hence, our findings indicate that *Ndrg2* is highly expressed in DC progenitor cells and promotes DC differentiation toward mature phenotypes. In addition to a reduction of global *Ndrg2* expression (Supplementary Fig. 4a), Ndrg2-KO using our Cas9-RNP approach almost eliminated cluster 9, which contained the Ndrg2+ progenitors and thereby halted DC maturation, preserving a premature DC state (Fig. 3f). These engineered premature DCs exhibited an even stronger *Vegfa* expression than DCs treated with VD3, and in addition exhibited an upregulation of extracellular matrix-associated genes that promote wound healing, such as *Fn1* (fibronectin-1) and *Mmp12* (matrix metalloproteinase-12), as well as the anti-oxidative genes *Prdx4* and *Mgst1* (Fig. 3g and Supplementary Fig. 4b). By contrast, control DCs highly expressed pro-inflammatory markers, such as *S100a6* and *S100a4* (Fig. 3h)[37].

To investigate differential regulation of cellular signaling pathways between our conditions, we then used GeneTrail 3, a computational pipeline for over-representation analysis (ORA) of specific gene sets on a single-cell level, and found a significant upregulation of pathways related to angiogenesis, epithelial cell migration, and degradation of the ECM in Ndrg2-KO cells, indicating a beneficial impact on wound healing (Fig. 3i)[48].

When analyzing the expression levels of genes located at the aforementioned potential off-target sites, *Amy1, Ftsj1, Sema5a*, and *Minar1* (Table S3), no detectable expression among all groups was found for *Sema5a* and *Minar1. Ftsj1* and *Amy1* were expressed at very low levels in less than 6% resp. 0.06% of all cells without significant differences, confirming that our approach for gene editing is highly

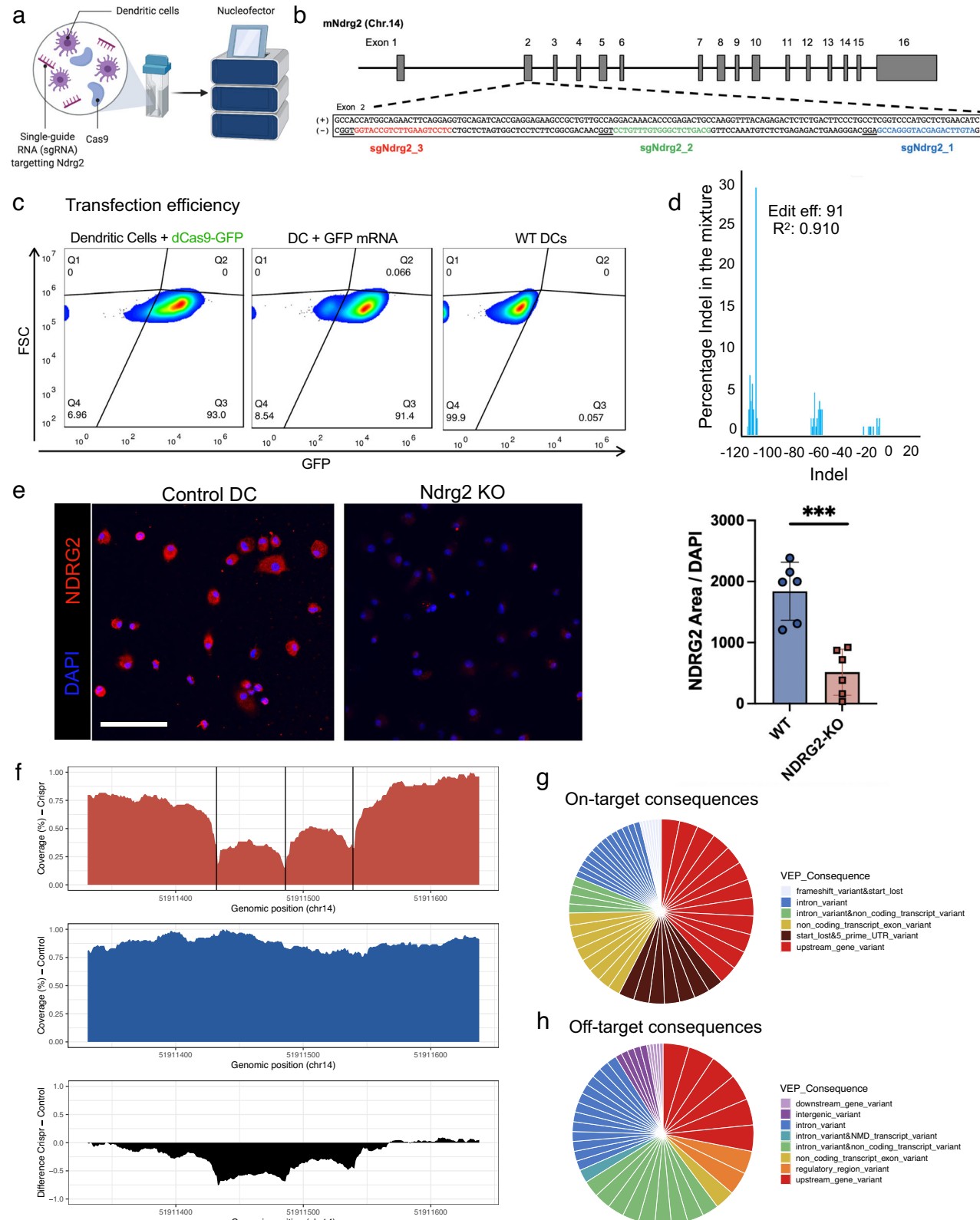

specific and does not cause significant off-target effects (Supplementary Fig. 3c).

## Hydrogel delivery of Ndrg2-KO DCs promotes healing of non-diabetic wounds

Having developed and characterized our approach for gene editing to specifically induce regenerative gene expression profiles in DCs, we then tested the suitability of these cells for application in a cell-based therapy for wound healing. The effectiveness and biocompatibility of hydrogels for cell delivery has been shown in murine and porcine wound models in combination with multiple cell types[5,49–53]. We seeded engineered DCs into a previously developed pullulan-collagen hydrogel to facilitate effective cell delivery onto the wound bed[49,53]. High viability and uniform distribution within the hydrogel were confirmed

**Fig. 2 | Development of a CRISPR/Cas9 platform for knockout of Ndrg2 in primary dendritic cells. a** Schematic for Cas9 ribonucleoprotein (RNP) nucleofection of primary dendritic cells (DCs). **b** Schematic of the Ndrg2 gene. Gray rectangles indicate exon regions. The sequences of the 3 single-guide RNAs (sgRNAs) targeting Ndrg2 (Ndrg2_1, Ndrg2_2, Ndrg2_3) are indicated in red, blue, and green. **c** Flow cytometry of untreated (wild-type, WT) DCs (negative control), DCs transfected with green fluorescent protein (GFP)-fused deactivated Cas9 (dCas9-GFP), and GFP-mRNA (positive control), indicating a high (93%) transfection efficiency of primary DCs with Cas9 protein. **d** Percentage of indels per given fragment size after Cas9 nucleofection of DCs. **e** Immunofluorescent staining for Ndrg2 in control and CRISPR-edited (Ndrg2-KO) cells (Student's *t*-test, unpaired and two-tailed: ***$P = 0.0003$, $n = 6$ biological replicates. Data are presented as mean values ± SEM.) Scale bar: 100 μm. **f** Coverage and reads generated by ultra-deep whole genome sequencing (100X) at the Ndrg2 on-target site for CRISPR-edited (red) and wild-type (blue) DCs. The difference in coverage between the groups is denoted in black. Vertical black lines indicate the 3 sgRNA cut sites. **g** Circle plots indicating different groups of on-target and **h** off-target consequences in CRISPR-edited cells (legend). White lines indicate the numbers of different consequences per group. Source data are provided as a Source Data file.

using calcein AM staining of live cells cultured on these hydrogels over the course of 21 days (Fig. 3j, Supplementary Fig. 6).

To determine the therapeutic potential of Ndrg2-KO-DCs, we treated splinted full-thickness excisional wounds in wild-type mice (C57BL6) with hydrogels seeded with either control DCs or Ndrg2-KO DCs. For controls, we subjected DCs to the same Cas9-RNP protocol, using 3 non-targeting sgRNAs to account for any potential effect of the Cas9-RNP or the nucleofection itself on wound healing. Hydrogels were placed onto the wound bed immediately after cell seeding. We also analyzed a second control group of wounds treated with blank, unseeded hydrogels (Fig. 4a). Wounds treated with our engineered Ndrg2-KO DCs healed significantly faster and were completely re-epithelialized by day 11, five days faster than in either control group (Fig. 4b–d). Histologically, healed wounds treated with Ndrg2-KO-DCs showed a fully regenerated epithelium and a thicker dermis compared to wounds treated with control DCs or blank hydrogels (Fig. 4c–e). Immunofluorescent staining for CD31 revealed a significantly stronger vascularization of healed wound tissue treated with engineered DCs compared to healed wound tissue from the control groups, confirming that the engineered DCs promoted angiogenesis and accelerated wound healing (Fig. 4f).

## Hydrogel delivery of Ndrg2-KO DCs promotes healing of diabetic wounds

We then investigated whether our DC therapy might also be beneficial for chronic wounds with impaired healing potential, such as in diabetes mellitus. Therefore, we treated excisional wounds in db/db mice with either Ndrg2-KO-DCs, control DCs (nucleofection with 3 non-targeting sgRNAs), or blank hydrogels ($n = 5$ per group, Fig. 4g). The db/db mice have a homozygous mutation in the leptin receptor gene (Lepr), resulting in morbid obesity, chronic hyperglycemia, and pancreatic beta cell atrophy. It has been shown that excisional wounds in db/db mice show a significant delay in wound closure, decreased granulation tissue formation, severely impaired vascularization, and reduced cell proliferation compared to wounds in WT and other diabetic mouse strains[54]. In accordance with our findings in WT mice, Ndrg2-KO DCs significantly accelerated wound healing in db/db mice, leading to complete re-epithelialization by day 16, five days earlier than in the control groups (Fig. 4h, i, Supplementary Fig. 7a, b) and similar to what we had observed in WT mice. The engineered DCs also promoted angiogenesis, similar to the effects seen in the WT mouse experiments, significantly improving vascularization of diabetic wound tissue compared to control DCs and unseeded hydrogels (Fig. 4j). Additionally, we utilized immunofluorescent staining to confirm that *Ndrg2* expression was not significantly different between unwounded skin and wound tissue explanted 3 days after wounding in wild-type (WT) and diabetic mice, demonstrating that ectopic expression of *Ndrg2* within the wound bed did not influence our findings (Supplementary Fig. 7c).

## Ndrg2-KO DCs target wound fibroblasts via growth factor signaling

We next aimed to further elucidate the molecular mechanisms underlying the accelerated wound healing and increased neovascularization in response to engineered DC therapy. To interrogate how the transplanted DCs interact with different cell types of the wound healing environment and affect their individual transcriptional signatures, we performed scRNA-seq of wound tissue treated with Ndrg2-KO-DCs, control DCs, or blank hydrogels (Fig. 5a). Tissue was explanted from a second batch of mice ($n = 5$ per group) on day 10 after wounding, when the wound healing curves of the different groups began to diverge (Fig. 4d), indicating critical differences in wound biology. We performed scRNA-seq on 10,612 cells across the three conditions, which were identified as fibroblasts, myeloid cells, neutrophils, lymphoid cells, and erythrocytes (Fig. 5b, c). Fibroblasts had the highest number of differentially expressed genes (DEGs, FC > 0.5, $p < 0.05$) between the groups among all cell types, indicating that our engineered DC therapy had the strongest impact on fibroblast gene expression within the wound bed (Fig. 5d, e, Supplementary Fig. 8a). Several genes that have been shown to be critically involved in wound healing were among the most differentially expressed genes in fibroblasts. *Ngfr*, the receptor for nerve growth factor, which has been shown to promote cell proliferation, angiogenesis, and wound healing, was almost exclusively expressed in fibroblasts from wounds treated with Ndrg2-KO DCs.(Fig. 5f, g)[55]. Moreover, treatment with Ndrg2-KO DCs strongly upregulated the expression of *Lgals1* (encoding galectin-1) in fibroblasts, which promotes wound healing in diabetic and non-diabetic wounds[56,57]. Interestingly, we also found an upregulation of *Plod2*, encoding lysyl hydroxylase 2, which catalyzes the first step of collagen crosslinking, and *Ppib*, encoding peptidyl prolyl cis/trans isomerase, which catalyzes the folding of ECM proteins such as collagen. These findings indicate that wound fibroblasts may be activated by Ndrg2-KO DCs to increase ECM protein synthesis and collagen crosslinking in the proliferative phase of wound healing, leading to accelerated wound closure.

To study the intercellular communication networks between the transplanted DCs and the local cells of the wound, we integrated scRNA-seq data from Ndrg2-KO and control DCs (Fig. 3a) with transcriptomic data from the respective local wound environments (Fig. 5b) and used the CellChat algorithm to analyze ligand-receptor interactions[58]. We found that Ndrg2-KO DCs showed a stronger outgoing signaling activity within their wound environment compared to control DCs (Fig. 5h) and also demonstrated a stronger total interaction strength within the aggregated cell-cell communication network (Fig. 5i). CellChat analysis demonstrated significant VEGF signaling from Ndrg2-KO DCs to fibroblasts, which was not present in control DCs, confirming our previous observations (Fig. 5j). In contrast to control DCs, Ndrg2-KO DCs also communicated with wound fibroblasts via other growth factor pathways, such as the insulin-like growth factor (IGF) and platelet-derived growth factor (PDGF) pathways (Fig. 5l, m), which stimulate cell proliferation and angiogenesis, and showed an upregulation of Spp1 (osteopontin) signaling, which has been shown to be a critical driver of early wound repair in diabetic wounds (Fig. 5m)[59–61].

## Discussion

Here, we have applied DC-based immunotherapy to complex diabetic and non-diabetic wounds and developed an effective gene

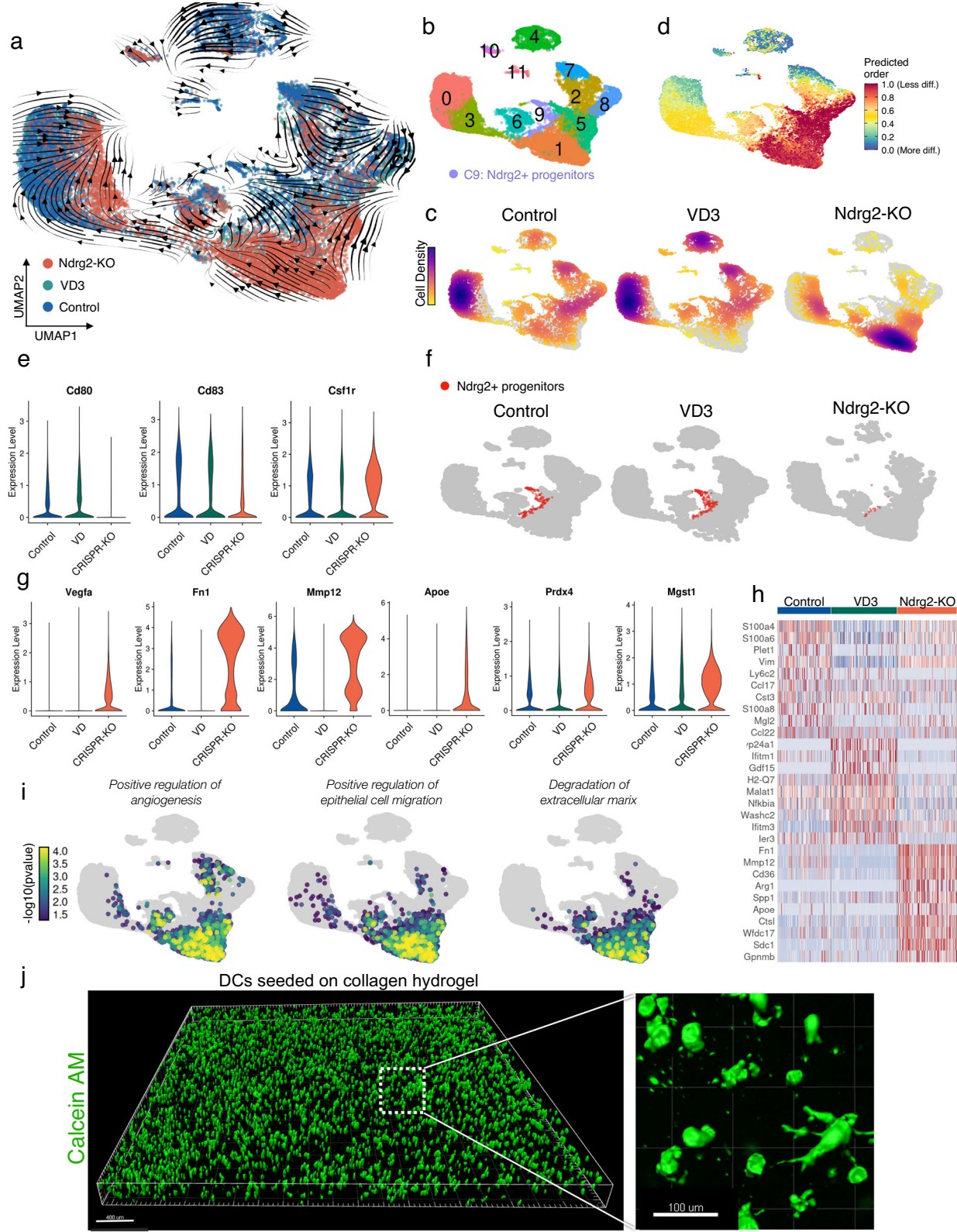

**Nature Communications** | (2023)14:4729

editing approach to potentiate its therapeutic efficacy. While stem cells have been widely investigated in pre-clinical studies as cell-based therapies for the treatment of wounds, the translational advancements many have hoped for have not been realized over the past decades[50,51]. By contrast, adoptive cell transfer using a patient's own T cells or DCs has been successfully translated to the clinic and has led to oncologic breakthroughs in recent years[62]. Combining gene editing with immunotherapy enabled the development of CAR-T-cell therapy, which has demonstrated unsurpassed effectivity against hematologic malignancies, even in patients who are refractory to multiple lines of therapy[63]. In addition to T cells, using a patient's own DCs to treat solid tumors and a variety autoimmune conditions has demonstrated adequate safety profiles and strong therapeutic efficacy[11,13,15,17,18].

**Fig. 3 | Ndrg2-knockout prevents dendritic cells maturation and induces regenerative gene expression profiles. a** UMAP embedding of single-cell RNA sequencing (scRNA-seq) data from Ndrg2-KO dendritic cells (DCs), vitamin D3 (VD3) stimulated DCs, and untreated cells. RNA velocity stream vectors computed with scVelo are projected onto the embedding. Cells colored by experimental group (23,871 cells). **b** Cells colored by Seurat cluster. Cluster 9 represents Ndrg2+ progenitor cells. **c** UMAP embedding split by experimental condition, and cells colored by cell density. In each plot, cells of the indicated condition are colored in yellow/violet and cells of the other two conditions are colored in gray. **d** UMAP embedding colored by CytoTRACE score indicating cell differentiation states; diff. = differentiated. **e** Violin plots of DC maturation markers. **f** UMAP embedding split by experimental condition. Ndrg2+ progenitor cells (cluster 9) colored in red. **g** Violin plots of wound healing-related markers. **h** Heatmap of the top 10 differentially expressed genes per cluster, sorted by average log fold-change and ordered by experimental condition. **i** Overrepresentation analysis (ORA) of the indicated gene sets (GO-BP) projected on the UMAP embedding. **j** Ndrg2-KO cells seeded on collagen-pullulan hydrogels stained with calcein AM. 3D reconstruction of stacked confocal microscopy images. Scale bars: 400 μm in overview, 100 μm in magnified image.

Gene editing of DCs for non-therapeutic use has previously been performed using lentiviral vector-based approaches. This technique, however, has several disadvantages for the development of clinical-grade DCs for cell-based therapies. Lentiviral transduction of DCs itself presents considerable challenges compared to other cell types. As professional antigen-presenting cells (APCs), DCs naturally express viral restriction factors, such as SAM domain and HD domain-containing protein 1 (SAMHD1), which allow them to deplete the cytoplasmic pool of deoxynucleoside triphosphates (dNTPs) necessary for the reverse transcription of viral genomes[64]. Hence, previous studies have reported relatively low transduction efficiencies (<40%) or required very high vector doses, which can induce undesired DC maturation or even cell toxicity[65]. Approaches to induce proteasomal degradation of SAMHD1 using Vpx or shRNA-mediated down-regulating of SAMHD1 to improve transduction efficiency have been described; however, these techniques are not suitable for the development of clinical-grade DC therapies since they increase the risk for cell infection with viruses and induce a cytotoxic T-cell response[66,67].

Recently, Cas9-RNP-based gene editing of CAR-T cells has been utilized for enhanced solid tumor therapy[22]. Furthermore, despite being early in the clinical test stage, Cas9-RNP-editing of murine primary DCs has been demonstrated to be an effective approach leading to high KO rates for several target genes, while preserving cell viability and in vitro functional characteristics[42]. Unlike vector-based approaches, Cas9-RNP-editing does not require the cellular transcription and translation of Cas9 to generate functional Cas9-sgRNA complexes, thus enabling higher peak Cas9 expression after transfection. In addition, rapid protein degradation of Cas9 from cells has been shown to increase CRISPR specificity and reduce the exposure of cells to Cas9, thus minimizing off-target effects[43].

Our approach for KO of *Ndrg2* in premature DCs using nucleofection with triple-guide Cas9-RNPs allows for streamlined, robust, and high-throughput editing of large quantities of DCs without the need for viral vectors. Our platform for gene editing consistently achieved >90% KO rates in primary DCs. To our knowledge, our study is the first to comprehensively assess off-target effects after Cas9-RNP nucleofection of DCs via ultra-deep DNA sequencing, demonstrating a high specificity of our CRISPR approach without detectable off-target effects.

Previous studies reported that *Ndrg2* is involved in DC maturation, however, its exact role has been poorly defined so far[33]. Using scRNA-seq of over 20,000 BM-DCs, we show that *Ndrg2* marks MDP/CDPs and that its knockout prevents DC maturation, preserving an immature cell state that is characterized by strong regenerative and pro-angiogenic gene expression profiles that cannot be achieved by pharmacologic treatment with VD3. In addition, targeted alteration of cells by CRISPR/Cas9 gene editing has a higher specificity and avoids pleiotropic effects of pharmacologic cell treatment. Moreover, CRISPR-KO provides a durable effect on cells, since target genes are eliminated, which may be of particular importance for future translational applications when genetically edited cell products may be stored as an 'off-the-shelf' therapy, or if clinical application of engineered cells must be delayed for other clinical considerations.

DCs edited with our approach were suitable for seeding in therapeutic hydrogels for in vivo application as a cell-based therapy. The collagen-pullulan hydrogels used in our study provide an optimal environment similar to native ECM and have been demonstrated to be an ideal vehicle for efficient cell delivery[49,53]. These hydrogels, even without cells, have a positive impact on wound healing[5,68]. In addition to delivery of CRISPR-edited cells for the treatment of chronic wounds, in vivo gene editing approaches have been developed that enable the local or systemic delivery of RNP to treat a variety of pathologic conditions[69]. Both viral and non-viral vectors have been used for therapeutic gene editing in the liver to treat conditions such as hemophilia, and phenylketonuria[70,71]. CRISPR/Cas9 gene editing can serve as an effective tool to treat various malignancies and has been successfully used to inhibit mechanisms of tumor growth. As gene editing approaches are typically used in advanced and often metastatic tumor stages, systemic injection is typically the preferred route of administration as it enables targeting lesions in multiple organ systems[72]. Systemic and local injection of viral vectors and extracellular vesicles have been used to perform CRISPR/Cas9 editing in muscular tissues[73]. Beyond that, numerous applications of CRISPR technologies for ocular conditions exist that predominantly utilize local injection into the eye, since smaller doses are sufficiently efficacious compared to systemic conditions where larger doses are necessary[69,74]. To our knowledge, our study is the first to describe the local delivery of CRISPR/Cas9 edited cells via hydrogels to improve the healing of chronic wounds in diabetic and non-diabetic models.

Wounds treated with control DCs did not heal faster compared to those treated with unseeded hydrogels, which indicates that DCs alone may not provide an additional benefit over collagen-pullulan hydrogels for wound healing. However, KO of Ndrg2 induced regenerative transcriptomic profiles in DCs, which led to significantly accelerated wound healing of WT as well as diabetic wounds by 5 days, surpassing what has previously been reported for genetically edited MSCs[24]. The improved vascularization of the wound bed after treatment with Ndrg2-KO DCs may be related to VEGF secreted from Ndrg2-KO DCs, which showed a much stronger expression of *Vegfa* compared to VD treated and control DCs. In addition, we found that edited DCs specifically directed growth factor signaling (VEGF, PDGF, and IGF) toward fibroblasts in the wound bed when analyzing the cellular ecology of the healing wounds in the proliferative phase of wound healing at day 10. Fibroblasts from wounds treated with edited DCs showed an upregulation of wound healing and ECM related genes, which indicates their strong therapeutic response to the secreted factors and provides a molecular basis for the observed accelerated wound closure with edited DCs.

A recent study demonstrated high editing efficiencies of human DCs with Cas9-RNP nucleofection[41], which indicates that our platform for gene editing may be translated to human DCs. Since clinical-grade manufacturing of autologous DCs for adoptive cell transfer is already well established in clinical practice and DC immunotherapy is currently being used in multiple clinical trials[18,75], we believe that genetically edited DCs have a high potential for successful clinical translation as a cell-based therapy for patients suffering from complex wounds that are refractive to current treatments. For future clinical applications,

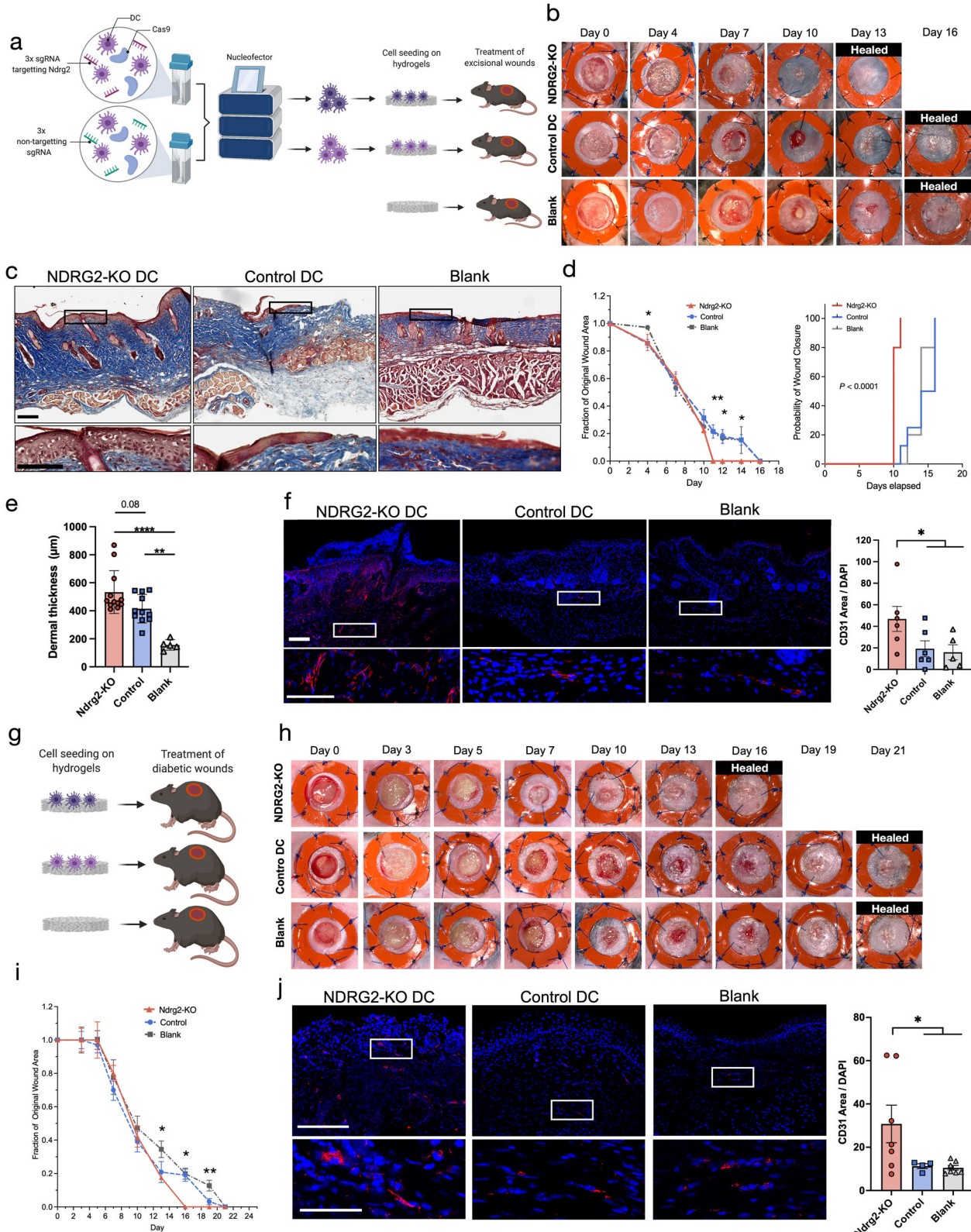

DC harvest could easily be performed from standard blood samples obtained at wound clinics, without the need for tissue biopsies or other surgical interventions (Supplementary Fig. 8b).

Our study has several limitations that have to be addressed in future studies before clinical translation can be achieved. Rodent models as employed in our study do not perfectly recapitulate human wound healing. Therefore, our findings should next be validated in a

large animal models, such as a porcine wound model, since porcine skin more closely mimics the biomechanics and physiology of human skin[52,76]. Future studies should also test different dosages of cells seeded onto the hydrogels to maximize the therapeutic potential of our technology for wound healing. Efforts have been made to optimize the specificity of CRISPR/Cas gene editing through protein engineering of Cas9 to increase its specificity or adoption of other Cas molecules (e.g.

**Fig. 4 | Hydrogel delivery of Ndrg2-KO dendritic cells promotes healing of diabetic and non-diabetic wounds. a** Schematic of cell seeding experiments on excisional wounds in wild-type mice (*n* = 5 per group). **b** Wounds treated with Ndrg2-KO dendritic cells (DCs) healed significantly faster and were completely re-epithelialized on day 11, 5 days faster than wounds treated with control DCs (nucleofection with 3 non-targeting sgRNAs) or blank hydrogels. **c** Masson's tri-chrome staining of explanted wound tissue (day 16) of the three groups. Zoomed-in panels show the regenerated epidermis. **d** Left: Relative wound area over time (Two-way analysis of variance (ANOVA) with Tukey's multiple comparisons test: Day 4: *$P < 0.05$, day 11: **$P < 0.01$, day 12: *$P < 0.05$, day 14: *$P < 0.05$, Ndrg2-KO: *n* = 8 biological replicates, Control: *n* = 8 biological replicates, Blank: *n* = 10 biological replicates. Data are presented as mean values ± SEM.), Right: Probability of wound closure in the three groups (reverse Kaplan–Meier estimate, ****$P < 0.0001$, *n* = 5 biological replicates per group). **e** Comparison of dermal thickness of wound tissue from the three experimental groups. One-way analysis of variance (ANOVA) with Tukey's multiple comparisons test: ****$P < 0.0001$, **$P = 0.02$, n = 12 biological replicates for Ndrg2-KO, *n* = 11 biological replicates for Control, n = 5 biological replicates for Blank. Data are presented as mean values ± SEM. **f** Immunofluorescent

staining, and quantification of wound tissue for CD31 marking blood vessels. One-way analysis of variance (ANOVA) with Tukey's multiple comparisons test: *$P < 0.05$, Ndrg2-KO: *n* = 5 biological replicates, Control: n = 10 biological replicates, Blank: *n* = 8 biological replicates. Data are presented as mean values ± SEM. Scale bars: 200 μm in overview, 100 μm in magnified images. **g** Schematic of cell seeding experiments on excisional wounds in diabetic mice. **h** Wounds treated with Ndrg2-KO DCs healed significantly faster and were epithelialized on day 16, 5 days faster than wounds treated with control DCs or blank hydrogels. **i** Relative wound area over time in diabetic wounds. Two-way analysis of variance (ANOVA) with Tukey's multiple comparisons test: Day 13: *$P < 0.05$, day 16: *$P = 0.05$, day 19: **$P = 0.01$, *n* = 5 biological replicates per group. Data are presented as mean values ± SEM. **j** Immunofluorescent staining, and quantification of diabetic wound tissue (day 21) for CD31. One-way analysis of variance (ANOVA) with Tukey's multiple comparisons test: *$P < 0.05$, *n* = 7 biological replicates for Ndrg2-KO, *n* = 4 biological replicates for Control, *n* = 7 biological replicates for Blank. Scale bars: 200 μm in overview, 100 μm in magnified images. Data are presented as mean values ± SEM. Source data are provided as a Source Data file.

Cas12a)[77,78]. Future studies should also test the efficacy of these Cas molecules for KO in DCs for the use of cell-based therapies.

In summary, our genetically edited DC therapy provides a highly effective, robust, and easy-to-use approach for the treatment of chronic wounds. This tunable therapy can be easily delivered to the wound bed using a hydrogel technology and has a strong potential for future translation into clinical trials, providing significant therapeutic opportunities for patients with diabetic and non-diabetic ulcers.

## Methods

All experiments were performed in accordance with Stanford University Institutional Animal Care and Use Committees and the NIH Guide for the Care and Use of Laboratory Animals. The study was approved by the Administrative Panel on Laboratory Animal Care (APLAC) at Stanford University (APLAC protocol number: APB-2965-GG0619).

### Isolation and cultivation of BM-DCs
Cultivation of DCs from the murine bone marrow was performed according to previously published protocols by Lutz et al.[25]. Briefly, after euthanasia, femurs and tibias were excised and the epiphyses on both ends of the long bones were removed. Bone marrow was collected by flushing the bones with 5 ml RPMI. Red blood cells lysis was performed with ACK lysis buffer. Cells were strained and then washed in complete media. Cells were cultured at a concentration of $2 \times 10^6$ per 100 mm culture dish in 10 ml RPMI containing 10% fetal bovine serum (FBS), 100 U/ml penicillin, 0.1 mg/ml streptomycin, 2 mM glutamine, 50 μM 2-mercaptoethanol, and 200 murine granulocyte-macrophage colony-stimulating factor (GM-CSF, Peprotech, Frankfurt, Germany). On day 3, 10 ml complete media were added to the culture dishes and on day 6, 10 ml of media were collected, cells were spun down and re-suspended in fresh media. For all experiments, only cells in suspension that represented the DC fraction of the cultures were used, and adherent cells were discarded. For in vitro experiments, DCs were treated on day 7 of culture for 24 h with either $10^{-6}$M 1,25 dihydroxy-cholecalciferol (Sigma-Aldrich, St. Louis, MO), $10^{-6}$M 1,25 dihydroxy-cholecalciferol and 100 ng/ml LPS (*Escherichia coli*, 026:B6; ThermoFisher) or $10^{-7}$ M dexamethasone (Sigma-Aldrich).

### Endothelial cell tube formation assay
Human umbilical vein endothelial cells (HUVECs) were obtained from Cell Applications (San Diego, CA). Tube formation assays were performed according to previously published protocols[79]. Briefly, HUVECs were seeded at $2 \times 10^5$ cells per 75 cm² in T75 culture flasks and expanded in Human EC Growth Medium (Cell Applications) until 75%

confluence was reached. After washing off growth media, cells were serum starved one day before the assay by incubation in Human EC Basal Medium (Cell Applications) for 24 h. Calcein AM (green; ThermoFisher) was added to cultures at a concentration of 2 μg/ml to stain cells and assess cell viability. Cells were incubated with Calcein AM in the dark at 37 °C and 5% $CO_2$ for 30 min, then the stain was washed off using PBS. Assays were performed in 24-well plates coated with Geltrex LDEV (lactose dehydrogenase elevating virus)-Free Reduced Growth Factor Basement Membrane Matrix (ThermoFisher) at 75 μl/cm² according to the manufacturer's recommendations. Cells were plated at a concentration of $3.5 \times 10^4$ in 200 μl Human EC Basal Medium (Cell Applications) per well. For co-culture assays, DCs were stained using CellTrace Calcein Red-Orange (ThermoFisher) and added to the EC cultures at a 1:1 ratio. Assays were run for 12 h and then fixed using 4% paraformaldehyde and imaged using a Zeiss LSM 880 confocal laser scanning microscope.

### CRISPR/Cas9-mediated Ndrg2 knockout and efficiency analysis
Single-guide RNA (sgRNA) design was performed using design tools from Integrated DNA Technologies (IDT, Newark, NJ) and Synthego (Menlo Park, CA). All spacer sequences of Ndrg2 sgRNA, negative control sgRNAs, primers for PCR amplification and DNA Sanger sequencing are listed in Table S2. RNPs were assembled using Cas9 Nuclease V3 purchased from IDT and directly synthesized together with Alt-R CRISPR-Cas9 sgRNAs from IDT or three guides as part of a multi-guide design from Synthego. For nucleofection we used the 4D-Nucleofector System and P3 Primary Cell 4D-Nucleofector X Kit L (Lonza, Basel, Switzerland). We optimized the RNP electroporation for DCs by using multiple-guide mixtures and different numbers of cells in a 100 μL total volume. The electroporated cells were plated in Costar Ultra-Low Attachmentat six-well plates (Corning) at 0.5 million cells per well in 2 mL medium. The cells from each well were harvested after 48 h and the total genomic DNA was extracted using Qiagen DNeasy Blood & Tissue kit. To analyze and verify KO efficiency, we amplified about 350 bp genome region covering 175 bp upstream and downstream of sgRNA targeting sites. The PCR products were column purified and then sequenced by Sanger Sequencing. The cutting and knockout efficiency were estimated using the ICE analysis tool (Synthego). After several runs of optimization, we used Cas9-RNP and the Lonza 4D-Nucleofector System together with the P4 Primary Cell 4D-Nucleofector X Kit L (Lonza) to transfect and create Ndrg2-KO DCs. To generate RNP, we used 18.6 pmol of Cas9 and 22 pmol of equimolar mixture of sgNDRG2_1, sgNDRG2_2 and sgNDRG2_3 (Synthego, Table S2, Supplementary Fig. 2a) per 0.5 million cells that were washed with DPBS and suspended in R Buffer (Lonza). After optimization, we processed 3 million DCs in each

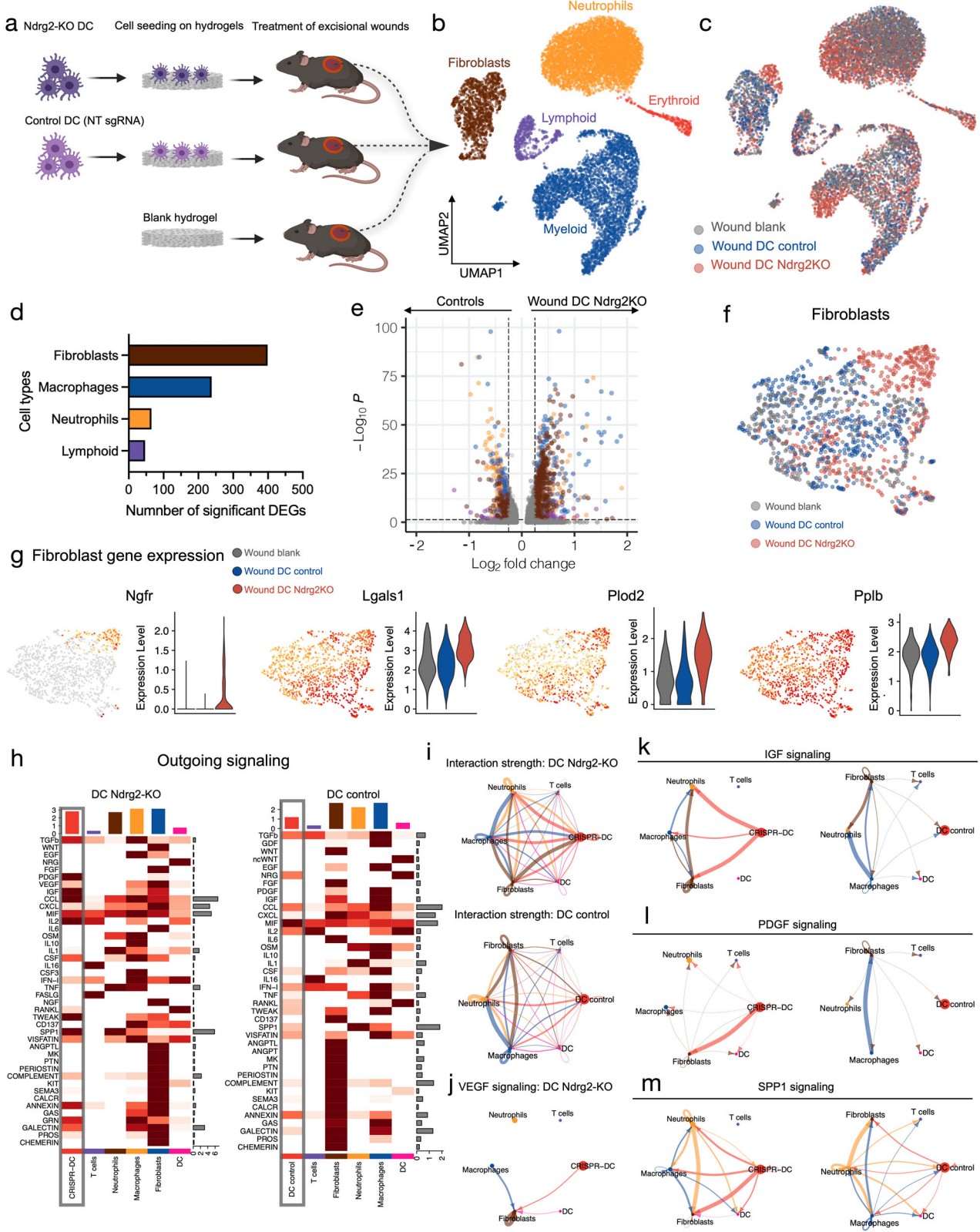

100 μL electroporation system for in vitro and in vivo experiments. All other steps followed IDT (Newark, NJ) Alt-CRISPR-Cas9 protocol. Cells were transfected using the program DK-100. We used a mixture of non-targeting sgRNAs (sgNC1, sgNC2, and sgNC3, Table S2) in the control RNP electroporation. For each cell production for in vivo experiments, we processed 30 millions of each Ndrg2-KO DCs and non-targeting control DCs.

**Whole genome sequencing**

Genomic DNA from Ndrg2-KO cells and control electroporated cells was extracted and whole genome sequencing (WGS) at ultrahigh depth (100X) was performed to confirm KO efficiency and evaluate off-target effects. WGS was performed by Novogene Corporation Inc. (Scaramento, CA). The genomic DNA was randomly fragmented by sonication to the size of 350 bp, then DNA fragments were end polished,

**Fig. 5 | Ndrg2-KO dendritic cells target wound fibroblasts via growth factor signaling. a** Schematic of cell seeding experiments to analyze wound tissue using single-cell RNA sequencing (scRNA-seq). **b** UMAP embedding of cells from all three conditions, colored by cell type. **c** Cells colored by experimental conditions. **d** Numbers of differentially expressed genes (DEG) colored by cell type (adjusted *p*-value < 0.05, FC > 0.5). Differentially expressed genes were determined using a Wilcoxon Rank Sum test and *P*-values were adjusted using the Benjamini−Hochberg procedure as part of the Seurat package. **e** Volcano plot showing significantly deregulated genes across all cell types. Right half of plot shows cells isolated from wounds treated with Ndrg2-KO dendritic cells (DCs); left half of plot shows cells isolated from control wounds; cells are colored by cell type as in **b** and **d. f** Subset of fibroblasts for all three conditions. **g** Expression of wound healing-related genes projected onto fibroblast subset UMAP embedding and as violin plots. **h** Heatmaps showing the expression of outgoing signaling pathways in Ndrg2-KO DCs (left) and control DCs (right) that target wound cells (integrated scRNA-seq data of Figs. 3 and 5). **i** Interaction strength with wound cells across differentially regulated signaling pathways for Ndrg2-KO DCs (top) and control cells (bottom). **j** VEGF signaling between Ndrg2-KO DCs and fibroblasts (not significant for control DCs). **k** IGF signaling in wounds treated with Ndrg2-KO DCs (left) or control DCs (right). **l** PDGF signaling in wounds treated with Ndrg2-KO DCs (left) or control DCs (right). **m** SPP1 signaling in wounds treated with Ndrg2-KO DCs (left) or control DCs (right).

A-tailed, and ligated with the full-length adapters of Illumina sequencing, and followed by further PCR amplification. The PCR products as the final construction of the libraries were purified with AMPure XP system. Then libraries were checked for size distribution by Agilent 2100 Bioanalyzer (Agilent Technologies, Santa Clara, CA), and quantified by real-time PCR (to meet the criteria of 3 nM). Samples were sequenced using Illumina Novaseq 6000 on an S4 Flowcell. Sequencing was performed according to the manufacturer's recommendations.

### Off-target analysis
The resulting reads for both samples were then aligned to the mouse reference genome (GRCm38) using Burrows-Wheeler Aligner (BWA, Version 0.7.15) in the 'bwa mem' setting with default parameters[80]. To identify potential off-target sites, we followed an approach of previous studies using CRISPR-edited cells for clinical applications in human patients[23]. First, we conducted variant calling using the strelka2 'somatic workflow' setting for paired samples (Version 2.9.2)[81]. All variants were then annotated using the Ensembl Variant Effect Predictor (Version 101, assembly = "GRCm38.p6", dbSNP = "150", gencode = "GENCODE M25"). Additionally, we used the Cas-OFFinder webtool (Version 2.4) to predict potential off-target sites using all three sgRNA sequences[45]. Here, we searched for sequences with five or fewer mismatches and DNA and RNA bulge sizes of two or less. To identify variants that might be potential off-target sites, we only considered insertion or deletion events that were not detected in the control sample, that passed all quality control checks ('FILTER = PASS'), and that did not contain the 'SNVHPOL' information in their VCF 'INFO' field. Additionally, we checked whether the remaining candidate sites are located within a 200 bp window up- or downstream of any predicted off-target site.

### Immunofluorescent staining and confocal laser scanning microscopy
After fixation, tissue was dehydrated in 30% sucrose in PBS for at least 48 h at 4 °C. Tissue was incubated in optimal cutting temperature compound (O.C.T., TissueTek, Sakura Finetek, Torrance, CA) for 24 h at 4 °C and cryoembedded in tissue molds on dry ice. Frozen sections were performed at 7 µm thickness on a cryostat. Antigen retrieval was performed using 0.01 M sodium citrate buffer in PBS (Abcam, Cambridge, MA), followed by blocking for 2 h in 5% goat serum (Invitrogen, Waltham, MA) in PBS. Sections were then incubated in anti-CD31 antibody (ab28364; Abcam, Waltham, MA) or anti-Ndrg2 antibody (ab174850; Abcam, Waltham, MA) at a 1:100 dilution overnight at 4 °C. Secondary antibodies were applied for 1 h at room temperature (Goat anti-Rabbit IgG (H + L) Highly Cross-Adsorbed Secondary Antibody, Alexa Fluor Plus 647; Thermo Fisher). For staining of cultured DCs, cells in suspension were collected from the culture dishes, spun down, and washed in PBS. Permeabilization was performed using 0.1% Triton X-100 in 1× PBS and cells were incubated at room temperature for 15 min. Cells were again washed and spun down, and blocking was performed using 1% BSA in 1× PBS for 1 h. Cells were incubated in recombinant Anti-NDRG2 antibody (ab174850; Abcam) in 500 µL of 0.1% BSA for 3 h, then washed three times in 1X PBS. Goat anti-Rabbit

IgG (H + L) Highly Cross-Adsorbed Secondary Antibody (Alexa Fluor Plus 647, Thermo Fisher) diluted in 500 µL of 0.1% BSA was applied for 45 min. Cells were again washed three times in 1X PBS, then mounted on slides using coverslips. Imaging was performed on a Zeiss LSM 880 confocal laser scanning microscope at the Cell Science and Imaging Facility at Stanford University. To obtain high resolution images, multiple serial images of ×25 magnification were acquired using automatic tile scanning. Individual images were stitched together during acquisition using the ZEN Black software (Zeiss, Oberkochen, Germany).

### Quantification of immunofluorescent staining
Immunofluorescent staining was quantified using a code written in MATLAB adapted from previous image analysis studies by one of the authors (K.C.)[52,82]. Briefly, confocal images were separated into their RGB channels and converted to binary to determine the area covered by each color channel. DAPI stain, corresponding to the blue channel, was converted to binary using *imbinarize* and an image-specific, automated threshold determined from the function *graythresh* in order to optimize the number of DAPI cells counted within the image. For the red and green channels, which corresponded to the actual protein stains, we converted these images to binary with a set a consistent threshold of ~0.3 for all images of the same stain. This consistent threshold ensured that our automated quantification of stain area would be unbiased. The area of red and green stains was then normalized by dividing by the number of cells, which we calculated as the number of DAPI nuclei above size threshold of 15 pixels. 3D reconstruction of confocal microscopy images was performed using Imaris 7.2 (Oxford Instruments, Abingdon, UK).

### Flow cytometry
Flow cytometry for GFP was performed on a BD FACS Aria (Becton Dickinson, San Jose, CA) and data was analyzed using FlowJo (Becton Dickinson, San Jose, CA).

### Single-cell RNA-seq data processing, normalization, and cell cluster identification
Base calls were converted to reads using the Cell Ranger (10X Genomics, Pleasanton, CA, USA; version 3.1) implementation *mkfastq* and then aligned against the mm10 (mouse) genome using Cell Ranger's count function with SC3Pv3 chemistry and 5000 expected cells per sample[52,53,76]. Cell barcodes representative of quality cells were delineated from barcodes of apoptotic cells or background RNA based on a threshold of having at least 300 unique transcripts profiled, less than 100,000 total transcripts, and less than 10% of their transcriptome of mitochondrial origin. Unique molecular identifiers (UMIs) from each cell barcode were retained for all downstream analyses. Raw UMI counts were normalized with a scale factor of 10,000 UMIs per cell and subsequently natural log transformed with a pseudocount of 1 using the R package Seurat (version 3.1.1)[83]. Aggregated data were then evaluated using uniform manifold approximation and projection (UMAP) analysis over the first 15 principal components[84]. Cell annotations were ascribed using the SingleR package (version 3.11) against

the ImmGen database[85,86] and confirmed by expression analysis of specific cell-type markers. Louvain clustering was performed using a resolution of 0.5 and 15 nearest neighbors.

## Generation of subpopulation markers, over-representation analysis, and cell-cell communication analysis

Cell-type markers were generated using Seurat's native *FindMarkers* function with a log fold-change threshold of 0.25 using the ROC to assign predictive power to each gene. Using GeneTrail 3 an ORA was performed for each cell using the 500 most expressed protein coding genes on the gene sets of the Gene Ontology[48]. P-values were adjusted using the Benjamini–Hochberg procedure and gene sets were required to have between 2 and 1000 genes. To predict intercellular communication between transplanted DCs and local wound cells, scRNA-seq data from Ndrg2-KO DCs and control DCs (Fig. 3a) were integrated with data from their specific wound environments (Fig. 5a) using Seurat. The CellChat package was used to analyze cellular communication networks using the "Secreted signaling" database[58].

## Statistical Analysis

Data are shown as mean ± standard error of the mean (SEM) and were analyzed using Prism 8 (GraphPad, La Jolla, CA). Two-group comparisons were performed with Student's *t*-test (unpaired and two-tailed). One- or two-way analysis of variance (ANOVA), followed by Tukey's post-hoc test were performed for comparisons of >2 groups as appropriate. $P < 0.05$ was considered statistically significant. The statistical methods used for scRNA-seq analysis are described in the specific sections above.

## Animals

Animals were housed under standard conditions according to protocols of the Stanford University Institutional Animal Care and Use Committees in a 24-h dark/light cycle and a temperature-controlled environment with access to food and water ad libitum. C57BL/6J (wild-type) and C57/BL/6-db/db mice were obtained from the Jackson Laboratory (Bar Harbor, ME). Female and male mice were randomly assigned to the different treatment groups. All animal surgeries were performed on 6-week-old animals under inhalation anesthesia with isoflurane (Henry Schein Animal Health) at a concentration of 1–2% in oxygen at 3 L/min. The mice were placed in prone position, and dorsal fur was shaved. Skin was disinfected with betadine solution followed by 70% ethanol three times.

## Cell seeding on hydrogels

Cells in suspension were collected from cultures and seeded on pullulan-collagen hydrogels with a diameter of 8 mm at a concentration of 500,000 cells per hydrogel and wound, using a previously described capillary force seeding technique[49]. We have previously demonstrated that this approach preserves cell viability as well as hydrogel architecture and enables efficient cell delivery[49,53].

## Splinted excisional wound model

Splinted full-thickness excisional wounds were created as previously described by Galiano et al. Here, a silicone ring is sutured around the wound margin to stent the skin, mimicking human physiological wound healing by limiting the contraction of the murine panniculus carnosus muscle and allowing the wounds to heal by granulation tissue formation and re-epithelialization[87]. Two full-thickness dermal wounds of 6 mm diameter were created on the dorsum of each mouse using biopsy punches. A silicone ring was fixed to the dorsal skin using an adhesive glue (Vetbond, 3M, Saint Paul, MN) as well as 8 interrupted 6-0 nylon sutures placed around the outer edge of the ring to prevent wound contraction. Collagen-pullulan hydrogels seeded with DCs or blank hydrogels. were placed onto the wound bed and the wounds were covered with sterile dressings (Tegaderm, 3 M, Saint Paul, MN).

Digital photographs were taken immediately after wounding and placement of the splints and during every dressing change until the time of wound closure. For the hydrogel treatment study, wounds were harvested on day 16 post wounding, when the wounds in all groups had healed.

## Reporting summary

Further information on research design is available in the Nature Portfolio Reporting Summary linked to this article.

## Data availability

Sequencing reads were aligned to the mouse reference genome (GRCm38). The authors declare that the source data supporting the findings of this study are provided with the manuscript and supplementary information files. The scRNA-seq data discussed in this publication have been deposited in NCBI's Gene Expression Omnibus88 and are accessible through GEO Series accession number GSE234145. Automated cell- level annotations were ascribed using the SingleR toolkit (version 3.11) against the ImmGen database. Source data are provided with this paper.

## Code availability

All other relevant code is available from the corresponding authors.

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

## Acknowledgements

This work was supported by the Plastic Surgery Research Foundation (Translational research grant 627063) and the German Research Foundation (HE 7621/1-1 (707175/808578)). We thank Yujin Park for her assistance in histopathology and Theresa Carlomagno for administrative support. Part of this work was performed at the Stanford Nano Shared Facilities (SNSF), supported by the National Science Foundation under award ECCS-1542152. Confocal imaging was performed at the Cell Sciences Imaging Facility, with generous support from the Beckman Foundation. Single-cell sequencing and qPCR was supported by the Stanford Functional Genomics Facility (SFGF) with funds from the NIH (S10OD018220 and 1S10OD021763). This work used the Genome Sequencing Service Center (GSSC) by Stanford Center for Genomics and Personalized Medicine Sequencing Center, supported by the grant award NIH S10OD025212, genome editing supported by the grant award NIH/NIA R21AG077193, and the Diabetes Genomics and Analysis Core of the Stanford Diabetes Research Center supported by grant award NIH/NIDDK P30DK116074. L.S.Q. is a Chan Zuckerberg Biohub – San Francisco Investigator. Figure schematics were created with BioRender.com.

## Author contributions

D.H., D.Z., D.S., L.S.Q. and G.C.G. designed the study. D.H., D.S., A.T., C.A.B., K.S.F., A.H.G., H.C.K., S.E.M.I. performed the animal experiments, A.H.G., S.E.M.I., J.P., J.A.B., K.C. performed in vivo animal data analysis, D.H., D.Z., D.S., A.T., C.A.B., K.C. performed in vitro experiments and analysis, D.H., K.C., T.K., M.J. and T.F. performed single-cell RNA sequencing data analysis, T.K. and H.P.L. performed whole genome sequencing data analysis, H.P.L., M.J., and A.K. performed and supervised bioinformatics analysis, D.H., D.Z., L.S.Q. and G.C.G. wrote the manuscript, U.K., B.L., M.T.L., K.C., L.S.Q. and G.C.G. helped revise and edit the manuscript. L.S.Q. and G.C.G. supervised all parts of the study.

## Competing interests

The authors declare no competing interests.

## Ethical approval

All experiments were performed in accordance with Stanford University Institutional Animal Care and Use Committees and the NIH Guide for the Care and Use of Laboratory Animals. The study was approved by the Administrative Panel on Laboratory Animal Care (APLAC) at Stanford University (APLAC protocol number: APB-2965-GG0619).
