## [Peer Review File · Nature Communications]

REVIEWER COMMENTS

Reviewer #1 (Remarks to the Author):

This is a very interesting manuscript taking advantage of CRISPR/Cas9 systems to achieve a high KO efficiency of the gene *Ndr2*, to ultimately enhance the regenerative potential of dendritic cells for wound healing. Interestingly, an hydrogel delivery system of optimised RNPs (3 gRNAs and Cas9) is implemented in vivo in diabetic mouse models. This work is exciting for the future translational studies of gene editing approaches in clinical trials for the treatment of wound healing conditions.

Major questions: The major issues of the manuscript have to do with specificity of the Cas9 KD. It would be important to assess the off targets using not just predictive (e.g. Cas9 OFFfinder) tools, but with more comprehensive and precise experimental mapping of OFF targets. In this line, how specific are these hydrogels in targeting DCs? Would it be necessary to target the Cas9 systems to achieve a cell-specific targeting in vivo? What are the biological effects of such hydrogels-RNPs delivery systems in other/non DC cell populations? Would ectopic expression of *Ndr2* be able to rescue the effect of the *Ndr2* KD? Have the authors studied the signaling pathways upon CRISPR-mediated KD by more direct techniques such as immunoblotting?

Minor issues/questions: The text has some issues, including word repetitions, and also editing mistakes and overall it would benefit from careful proof-reading of the text. Error bars in histograms do not contain individual data points.

It will be also important that the authors discuss other delivery systems currently available for Cas9/RNP delivery in the discussion and adding a more broad comparison with other available delivery tools for Cas9 technologies.

Reviewer #2 (Remarks to the Author):

The authors of the present manuscript have developed a CRISPR/Cas9 approach to precisely edit dendritic cells (DCs) to enhance their therapeutic potential for healing chronic wounds. Using single-cell RNA sequencing (scRNA-seq) of tolerogenic DCs, the authors reported the discovery of N-myc downregulated gene 2 (*Ndr2*), which marks a specific population of DC progenitors, as a target for CRISPR knockout (KO). The authors then incorporated the CRISPR-based cell engineering system within a hydrogel technology for in vivo cell delivery. They developed a translational approach for DC-based

immunotherapy that accelerated the healing full-thickness wounds in non-diabetic and diabetic mouse models. The data sounds but I am unable to confirm the ambitious claims based on the reports.

Reviewer #3 (Remarks to the Author):

In this manuscript, the authors declared that they provides an efficient, robust, and easy-to-use approach for the treatment of chronic wounds with gene-edited DC therapy. While this has important implications for the use of cell-based therapies for disease treatment, many issues are not clear. While the results are compelling, there are still some problems to be corrected in the whole paper.

Specific comments:

1. "Dendritic cells (DCs) play critical roles in regulating both innate and adaptive immune responses." Does the use of Cas9 gene edited DC cells for skin repair cause severe immune rejection and how to solve the problem?
2. "DCs can be easily isolated from the peripheral blood." But is it practical to obtain DCs from peripheral blood and then use gene editing techniques to process the DCs for acute skin treatment?
3. "Multiple clinical trials have reported beneficial effects of tolerogenic DC therapy in the treatment of autoimmune diseases, such as multiple sclerosis, rheumatoid arthritis, type 1 diabetes, and Crohn's disease, as well as transplant rejection". They are all immune-related diseases, so the use of tolerogenic DC to modulate the immune response for the treatment of these diseases is quite understandable. But why did the authors choose tolerogenic DC for skin repair treatment when skin repair is not strictly an immune-related disease?
4. The authors found that Ndr2 expression was downregulated after treatment of DCs with VD3 (figure 1C). Then they used VD3 and lipopolysaccharide (LPS) or dexamethasone treated DCs co-cultured with EC, and the results showed that Ndr2-inhibited DCs stimulated EC tube formation in vitro, as indicated by a higher EC branch and junction number, longer branch length, and higher total mesh area compared to EC co-cultures with untreated DCs and ECs cultured in basal media. The decrease in Ndr2 expression is only a phenomenon after treatment of DCs with drugs such as VD3, how the authors concluded that Ndr2 is the main cause of promoting vascularization of ECs, after all, they did not evaluate whether there would be differential expression of B or C genes affecting vascularization after VD3 treatment of DCs. In other words, could the Ndr2 gene be only a downstream phenomenon rather than a major factor in promoting the vascularization of ECs after VD3 treatment of DCs?

5. The off-target of CRISPR/Cas9 has been the subject of several rounds of controversy, and safety is one of the decisive factors for a new technology to be able to be used in the clinic. Although everyone's study declares that the Cas system they use has good targeting, the results are not satisfactory. Although the authors of this paper also used the commonly used Cas-OFFinder to detect its off-target efficiency (figure 2), but this belongs to the bioinformatics category, did the authors verify its off-target efficiency from the biological perspective?

6. The repaired tissue images should be given in figure 4b and 4h.

7. The author mentioned that "High viability and uniform distribution within the hydrogel were confirmed using calcein AM staining of live cells cultured on hydrogels for 7 days". Survival and bioactivity of engineered DCs following hydrogel delivery can determine their usage and effectiveness. Please perform calcein AM staining of live cells cultured on hydrogels at different time points. The maximum observation time should be 21 days as the authors mentioned in Figure 4h or no viable cells.

8. When the authors further elucidated the molecular mechanisms underlying the accelerated wound healing in response to gene-edited DC therapy, the authors mentioned that Ndr2-KO DCs target wound fibroblasts. It is unclear about the ECs and this mechanism in vivo.

9. Please compare the expression of Ndr2 before and after wounding in wild-type and diabetic mice.

10. In Figure 4, wounds treated with control DCs did not heal faster compared to those treated with unseeded hydrogels, which indicates that DCs alone may not provide an additional benefit over collagen-pullulan hydrogels for wound healing. Please discuss the reason in the Discussion section.

11. As a powerful and easy-to-use gene editing tool, CRISPR/Cas9 has great potential in scientific research fields. However, this gene editing technology is not perfect, and the off-target effects are a key factor limiting its clinical applications. The authors developed a novel approach for KO of Ndr2 in DCs, which minimizing off-target effects. However, there is no evidence that this novel approach is completely free of off-target effects. Please mention this in the Discussion section and discuss the other methods to reduce the off-target effects in the Discussion section with latest reference (e.g., <https://doi.org/10.1038/s41586-022-04470-1>).

12. Whether the system is safe for skin repair and its safety needs to be tested.

Reviewer #4 (Remarks to the Author):

The manuscript shows that deletion of *Ndr2* leads to improved wound healing by BM derived DC populations. The early part of the manuscript is confusing and makes it uncertain which cell populations are actually being affected to produce the effects on wound healing seen in the latter parts of the manuscript. The manuscript employs an impressive array of technologies and robust wound healing modeling. The effect on wounds healing is convincing.

In Figure 1, the authors show 9 clusters but then only describe 3 of them. This is confusing. Are they really distinct cell populations? How do they compare to previously described DC subpopulations? The protocol for preparing the DCs does not provide a purification step or step confirming the DC identity of the cells. The scRNA-seq provides that opportunity. The cells in clusters C4, C6, C7 and C9 (Figure 1) need to be identified. A supplemental Table showing the top differentially expressed genes/cluster should be provided.

The data provided do not convincingly show that cluster 8 contains the progenitor DC population. As a start the authors do not show that CD34 expression is limited to this population. The rationale for focusing on cluster 8 for the tolerogenic cells is unclear, other than they express higher levels of *Ndr2*.

In Fig 1 the authors do not show clustering by treatment except for cluster 8, making it impossible to understand the VD3 changes undergone in other clusters. Figure 3c shows indeed that VD3 cells are found in other clusters and appear to be present in higher proportion in 4 and 7 in addition to cluster 9 (=8 in Fig 1 UMAP). It would be helpful to show this earlier as the reader needs to understand this information associated with Figure 1.

For both the UMAP clustering in Figures 1 and 3, my tendency would be for the UMAP clusters to be either: 1) combined into the subsets the investigators are able to identify clearly, 2) adjust the sensitivity of the UMAP clustering to more accurately reflect the known subsets or 3) provide feature plots or violins showing the distinctive gene expression associated with the different clusters and mention that the implications of the different DC clusters is unknown.

Controls for the experiments in Figure 1i are lacking. What is the effect of LPS and VD3 directly on the endothelial cell cultures, or cultures with no DC and no treatment. The ECs shown at baseline appear very poorly formed.

Please label the axes of Figure 2c.

Altered gene expression by the fibroblasts should be made more granular. In the absence of keys in Fig 5e and 5h, I find these panels impossible to interpret. As a reader, I am anxious to know more broadly the genes upregulated in the *Ndr2* k/o DC-treated wounds. This could be provided by a table, and/or by a more complete labeling of the genes in 5e. In any event supplemental tables providing the DEGs for each cluster should be provided.

Response to reviewers

Nature Communications manuscript NCOMMS-22-24794

Title: Cas9-Mediated Knockout of *Ndrp2* Enhances the Regenerative Potential of Dendritic Cells for Wound Healing

Reviewer #1 (Remarks to the Author):

This is a very interesting manuscript taking advantage of CRISPR/Cas9 systems to achieve a high KO efficiency of the gene *Ndrp2*, to ultimately enhance the regenerative potential of dendritic cells for wound healing. Interestingly, an hydrogel delivery system of optimised RNPs (3 gRNAs and Cas9) is implemented in vivo in diabetic mouse models. This work is exciting for the future translational studies of gene editing approaches in clinical trials for the treatment of wound healing conditions.

- We thank the reviewer for their supportive and constructive comments.

Major questions: The major issues of the manuscript have to do with specificity of the Cas9 KD. It would be important to assess the off targets using not just predictive (e.g. Cas9 OFFfinder) tools, but with more comprehensive and precise experimental mapping of OFF targets. In this line, how specific are these hydrogels in targeting DCs? Would it be necessary to target the Cas9 systems to achieve a cell-specific targeting in vivo? What are the biological effects of such hydrogels-RNPs delivery systems in other/non DC cell populations? Would ectopic expression of *Ndrp2* be able to rescue the effect of the *Ndrp2* KD? Have the authors studied the signaling pathways upon CRISPR-mediated KD by more direct techniques such as immunoblotting?

- We thank the reviewer for their comment. We would like to clarify that we have not used hydrogels that deliver RNPs to target DCs as the reviewer describes. Instead, we have used Cas9-RNP technology to knock out *Ndrp2* in bone marrow-derived DCs in vitro. These engineered DCs were then seeded onto hydrogels for delivery of the cells onto wounds where they exert their regenerative effect to promote wound healing. We apologize if parts of our manuscript were not phrased clearly and have caused a misunderstanding.

With regard to the off-target analysis, we have used whole genome sequencing (WGS) with an ultrahigh depth (~100X coverage) of CRISPR-edited and unedited control DCs in addition to computational prediction with CasOFFinder. WGS provides an unbiased survey of the full genome for off-target nuclease activity and has been used in multiple prior studies on CRISPR Cas9 gene editing for therapeutic applications.^{1,2} Studies have reported that between 33X and 50X coverage is necessary to detect single-nucleotide polymorphisms in human genomes.^{2,3} In our study we performed WGS with an ultrahigh depth of 100X, which is twice the depth that has been recommended for off-target analysis in prior studies.^{2,4,5} Thus, we feel that our approach has a very high sensitivity to detect potential off-target sites. In our analysis, indels were not detected within 15 base pairs (bp) up- and downstream of the sgRNA binding sites. When each site was broadened to 200bp up- and down-stream, indels in 14 sites were identified (**Fig. 2h, Table S3**). By manual inspection of these loci, we found that only 7 loci showed indels that occurred only in the *Ndrp2*-KO group and not in the control group. After identifying the most likely off-target sites, we further aligned the sequence between the predicted Cas-OFFinder cut sites and

the three sgRNA sequences targeting *Ndr2*. Among the three sites that showed the highest alignment, we only identified a protospacer adjacent motive (PAM) in two of the sites. All other predicted off-target sites did not show similarity to the sgRNA sequences and therefore are unlikely true off-target sites (**Extended Data Fig. 3b, c**). This set of analysis together confirmed that the designed guide RNAs and the use of RNP in DCs cells had little or no observable off-target effects.

With regard to the biologic effects of our cell delivery technology on other cell types, we have used single-cell RNA sequencing (scRNA-seq) of wound tissue to determine how the engineered DCs impact different cell types involved in the wound healing process. scRNA-seq showed that the engineered DCs have the strongest impact on fibroblasts within the wound and upregulate regenerative transcriptomic profiles in these cells (Figure 5).

As the reviewer suggested, we have performed additional experiments to identify potential effects from ectopic *Ndr2* expression within the wound bed that could have confounded our results. To do this, we performed immunofluorescent staining for *Ndr2* on unwounded skin and wound tissue explanted on day 3 after wounding in wild-type and diabetic mice. In both mouse strains, we did not find significant differences in *Ndr2* expression between unwounded skin and wound tissue. These findings indicate that the regenerative effects on wound healing that we found after application of our DC hydrogels are not related to ectopic expression of *Ndr2* within the wound bed.

Minor issues/questions: The text have some issues, including word repetitions, and also editing mistakes and overall it would benefit from careful proof-reading of the text. Error bars in histograms do not contain individual data points.

- Thank you so much for pointing this out. We apologize for any editing errors in our manuscript. We have revised the manuscript to correct any typos and word repetitions. We have also included individual data points on the bar plots in the revised manuscript.

It will be also important there authors discuss other delivery systems currently available for Cas9/RNP delivery in the discussion and adding a more broad comparison with other available delivery tools for Cas9 technologies.

- Thank you so much for raising this important point. We have added a paragraph on other delivery systems of Cas9/RNP to the discussion of the revised manuscript:

In addition to delivery of CRISPR-edited cells for the treatment of chronic wounds, in vivo gene editing approaches have been developed that enable the local or systemic delivery of RNP to treat a variety of pathologic conditions.⁶ Both viral and non-viral vectors have been used for therapeutic gene editing in the liver to treat conditions such as hemophilia, and phenylketonuria.^{7,8} CRISPR/Cas9 gene editing can serve as an effective tool to treat various malignancies and has been successfully used to block tumor growth mechanisms. As novel gene editing approaches are typically used in advanced and often metastatic tumor stages, systemic injection is typically the preferred route of administration as it enables targeting lesions in multiple organ systems.⁹ Systemic and local injection of viral vectors and extracellular vesicles have been used to perform CRISPR/Cas9 editing in muscular tissues.¹⁰ Beyond that, numerous applications of CRISPR technologies for ocular conditions exist that mostly use local injection into the eye, since smaller doses are required than for systemic conditions.^{6,11} To our knowledge, our study is the first to

describe the local delivery of CRISPR/Cas9 edited cells via hydrogels to improve the healing of chronic wounds in diabetic and non-diabetic models.

Reviewer #2 (Remarks to the Author):

The authors of the present manuscript have developed a CRISPR/Cas9 approach to precisely edit dendritic cells (DCs) to enhance their therapeutic potential for healing chronic wounds. Using single-cell RNA sequencing (scRNA-seq) of tolerogenic DCs, the authors reported the discovery of N-myc downregulated gene 2 (Ndr2), which marks a specific population of DC progenitors, as a target for CRISPR knockout (KO). The authors then incorporated the CRISPR-based cell engineering system within a hydrogel technology for in vivo cell delivery. They developed a translational approach for DC-based immunotherapy that accelerated the healing full-thickness wounds in non-diabetic and diabetic mouse models. The data sounds but I am unable to confirm the ambitious claims based on the reports.

- We thank the reviewer for their comments on our manuscript. We fully acknowledge that our study has limitations and further studies will be required before clinical translation of our new technology can be achieved. Rodent models, as employed in our study, do not fully recapitulate human wound healing, and therefore, our findings need to be validated in porcine wound models in the future, since porcine skin more closely mimics the biomechanics and physiology of human skin. Future studies should also test different dosages of cells seeded onto the hydrogels to maximize the therapeutic potential of our technology for wound healing. We have added a paragraph on these study limitations to the revised version of our manuscript.

Reviewer #3 (Remarks to the Author):

In this manuscript, the authors declared that they provide an efficient, robust, and easy-to-use approach for the treatment of chronic wounds with gene-edited DC therapy. While this has important implications for the use of cell-based therapies for disease treatment, many issues are not clear. While the results are compelling, there are still some problems to be corrected in the whole paper.

Specific comments:

1. "Dendritic cells (DCs) play critical roles in regulating both innate and adaptive immune responses." Does the use of Cas9 gene edited DC cells for skin repair cause severe immune rejection and how to solve the problem?

- Thank you very much for this interesting question. In our study, we used dendritic cells (DCs) isolated from the bone marrow of inbred wild-type C57/BL6 mice for CRISPR/Cas9 gene editing. These edited DCs were then delivered to wounds on mice of the same highly inbred strain, which are genetically identical to each other. Thus, we do not expect any immune rejection of the cells. In future clinical applications of our technology, autologous DCs could be isolated from the peripheral blood of a patient according to established protocols and then transferred back to the patient's chronic wound after in vitro gene editing.

2. “DCs can be easily isolated from the peripheral blood.” But is it practical to obtain DCs from peripheral blood and then use gene editing techniques to process the DCs for acute skin treatment?

- Thank you very much for this comment. In the United States, more than 6.5 million patients suffer from chronic wounds, which impact nearly 15% of Medicare beneficiaries, costing the U.S. healthcare system \$28 - 32 billion annually.^{12,13} Clinically established standard approaches such as skin grafting or tissue engineered skin substitutes often require surgical interventions and frequently fail in the setting of complex wounds in patients with diabetes and peripheral vascular disease¹⁴. Novel cell-based therapies have demonstrated significant benefits for the treatment of chronic wounds with secretory, immunomodulatory, and regenerative effects.¹⁵ With regard to the practical implementation of our novel DC therapy, we do not expect major obstacles in the process of DC isolation from the peripheral blood to perform gene editing in vitro, followed by transferring the edited cells back into the patient. A similar pipeline has been clinically well established for processing of CAR T cells since 2017. So far, six CAR T cell therapeutics have been approved by the FDA for the treatment of hematologic malignancies and are being used in clinical practice. Since many of the processes of cell apheresis from the peripheral blood and gene editing have been widely established by the industry for CAR T cell production, we do not expect that gene editing of autologous DCs for the treatment of chronic wounds will create an obstacle with regard to manufacturing and clinical implementation.^{16,17}

3. “Multiple clinical trials have reported beneficial effects of tolerogenic DC therapy in the treatment of autoimmune diseases, such as multiple sclerosis, rheumatoid arthritis, type 1 diabetes, and Crohn’s disease, as well as transplant rejection”. They are all immune-related diseases, so the use of tolerogenic DC to modulate the immune response for the treatment of these diseases is quite understandable. But why did the authors choose tolerogenic DC for skin repair treatment when skin repair is not strictly an immune-related disease?

- Thank you very much for this important question. The innate immune system plays a critical role during wound repair by clearing damaged cells and initiating the tissue repair process, which eventually results in scar formation. Macrophages and dendritic cells (DCs) are central components of the innate immune response and are critically involved in all phases of wound healing, from initial inflammation to tissue remodeling.¹⁸⁻²⁰ While many studies have demonstrated the critical impact of distinct macrophage subpopulations for decades,^{19,21} several recent studies have pointed to a critical importance of DCs for wound healing in diabetic and non-diabetic conditions.²²⁻²⁴ For cell-based therapy approaches, DCs can be easily isolated from the peripheral blood in large quantities using established good manufacturing practice (GMP) protocols for leukaphereses.²⁵ In addition, they can be easily expanded in vitro using established protocols and are well suitable for seeding on cell delivery hydrogels that can be applied to the wound bed as we have shown in our study.²⁵ Therefore, we believe that DCs are the ideal cell type for cell-based therapies to treat chronic wounds. By combining our hydrogel technology for in vivo DC delivery with safe in vitro CRISPR/Cas9 cell engineering we have developed an effective translational approach for DC therapy of chronic wounds and hope to open the door for future clinical trials.

4. The authors found that *Ndr2* expression was downregulated after treatment of DCs with VD3 (figure 1C). Then they used VD3 and lipopolysaccharide (LPS) or dexamethasone treated DCs co-cultured with EC, and the results showed that *Ndr2*-inhibited DCs stimulated EC tube formation in vitro, as indicated by a higher EC branch and junction number, longer branch length, and higher total mesh area compared to EC co-cultures with untreated DCs and ECs cultured in basal media. The decrease in *Ndr2* expression is only a phenomenon after treatment of DCs with drugs such as VD3, how the authors concluded that *Ndr2* is the main cause of promoting vascularization of ECs, after all, they did not evaluate whether there would be differential expression of B or C genes affecting vascularization after VD3 treatment of DCs. In other words, could the *Ndr2* gene be only a downstream phenomenon rather than a major factor in promoting the vascularization of ECs after VD3 treatment of DCs?

- Thank you very much for this important comment. We completely agree with the reviewer that VD3 treatment transcriptionally affects a plethora of genes in DCs, some of which may have a beneficial effect on angiogenesis. Our reason for choosing *Ndr2* as a target for CRISPR-KO in our study was two-fold. First, we found that tolerogenic DCs induced by treatment with VD3 showed a downregulation of *Ndr2*. Previous studies have reported that *Ndr2* is a potent inhibitor of growth factor expression, angiogenesis, and cell proliferation.²⁶⁻³⁰ Therefore, *Ndr2* appeared to be an interesting target for further investigation. In addition, scRNA-seq of VD3 treated and control DCs demonstrated that *Ndr2* marks a specific progenitor population of MDP (macrophage and dendritic cell progenitor) and CDP (common dendritic cell progenitor). Hence, we wondered how KO of this marker would affect the transcriptional dynamics of DC maturation in our cultures. Using our Cas9 RNP KO strategy, we demonstrated that *Ndr2*-KO shifts DC populations to a more pre-mature state and induced regenerative and pro-angiogenic transcriptomic profiles. Using in vivo experiments on murine excisional wounds, we show that these edited cells are ideal candidates for cell-based therapies to promote healing of diabetic and non-diabetic wounds.

5. The off-target of CRISPR/Cas9 has been the subject of several rounds of controversy, and safety is one of the decisive factors for a new technology to be able to be used in the clinic. Although everyone's study declares that the Cas system they use has good targeting, the results are not satisfactory. Although the authors of this paper also used the commonly used Cas-OFFinder to detect its off-target efficiency (figure 2), but this belongs to the bioinformatics category, did the authors verify its off-target efficiency from the biological perspective?

- In addition to Cas Off-Finder, we used whole genome sequencing (WGS) with an ultrahigh depth (~100X coverage) of CRISPR-edited DCs and non-edited control DCs. WGS enables an unbiased survey of the entire genome for off-target nuclease activity and has been used in multiple prior studies on CRISPR Cas9 gene editing for therapeutic applications.^{1,2} Studies have reported that between 33X and 50X coverage is necessary to detect single-nucleotide polymorphisms in human genomes.^{2,3} In our study, we performed WGS with an ultrahigh depth of 100X on edited and non-edited DCs, which is twice the depth that has been recommended for off target analysis in prior studies.^{2,4,5} Thus, we feel that our approach has a very high sensitivity to detect potential off-target sites. In our analysis, indels were not detected within 15 base pairs (bp) up- and downstream of the sgRNA binding sites. When each site was broadened to 200bp up- and down-stream, indels in 14 sites were identified (**Fig. 2h, Table S3**). By manual inspection of these loci, we found that only 7 loci showed indels that occurred only in the *Ndr2*-KO group and not in the control group. After identifying the most likely off-target sites, we

further aligned the sequence between the predicted Cas-OFFinder cut sites and the three sgRNA sequences targeting *Ndrg2*. Among the three sites that showed the highest alignment, we only identified a protospacer adjacent motive (PAM) in two of the sites. All other predicted off-target sites did not show similarity to the sgRNA sequences and therefore are unlikely true off-target sites (**Extended Data Fig. 3b, c**). This set of analysis together confirmed that the designed guide RNAs and the use of RNP in DCs cells had little or no observable off-target effects.

6. The repaired tissue images should be given in figure 4b and 4h.

- We have added images showing the healed wounds to **Figures 4b and 4h**.

7. The author mentioned that “High viability and uniform distribution within the hydrogel were confirmed using calcein AM staining of live cells cultured on hydrogels for 7 days”. Survival and bioactivity of engineered DCs following hydrogel delivery can determine their usage and effectiveness. Please perform calcein AM staining of live cells cultured on hydrogels at different time points. The maximum observation time should be 21 days as the authors mentioned in Figure 4h or no viable cells.

- Thank you very much for this important comment. We have performed calcein AM staining and confocal microscopy of DCs seeded on hydrogels over the course of 21 days as the reviewer suggested. We found that the viable DC population on the hydrogels did not significantly change in abundance across this observation period (**Extended Data Fig. 6**). These findings confirm that DCs remain viable within the hydrogel environment for 21 days, further underscoring their suitability for cell-based therapy approaches to treat chronic wounds.

8. When the authors further elucidated the molecular mechanisms underlying the accelerated wound healing in response to gene-edited DC therapy, the authors mentioned that *Ndrg2*-KO DCs target wound fibroblasts. It is unclear about the ECs and this mechanism in vivo.

- Thank you very much for this valuable comment. We apologize that the interpretation our transcriptomic analysis regarding the biologic relationship of *Ndrg2*-KO DCs, fibroblasts and ECs was not clearly stated in our manuscript. Single-cell RNA sequencing revealed a significantly higher expression of *Vegfa* (encoding vascular endothelial growth factor) in *Ndrg2*-KO DCs compared to VD treated and control DCs. Therefore, we hypothesize that VEGF secreted from *Ndrg2*-KO DCs stimulates angiogenesis within the wound bed that promotes healing. In addition, our transcriptional analysis of wound tissue in Fig. 5 revealed the strongest interaction between *Ndrg2*-KO DCs and fibroblasts (**Figure 5**). When analyzing differentially expressed genes in fibroblasts from wounds that had been treated with *Ndrg2*-KO DCs vs. non-edited DCs, we identified several genes that are related to angiogenesis (**Figure 5g**). Fibroblasts from wounds treated with *Ndrg2*-KO DCs almost exclusively expressed *Ngfr*, the receptor for nerve growth factor, which has been shown to promote angiogenesis.³¹ Moreover, treatment with *Ndrg2*-KO DCs strongly upregulated *Lgals1* (encoding galectin-1) in fibroblasts, which also has been shown to promote cell proliferation and angiogenesis.^{32,33} Therefore, our data indicate that *Ndrg2*KO DCs directly stimulate angiogenesis during wound healing via VEGF and also promote a regenerative and pro-angiogenic fibroblast phenotype. We have added this information to the discussion section of our manuscript.

9. Please compare the expression of *Ndr2* before and after wounding in wild-type and diabetic mice.

- Thank you very much for this great suggestion. We have performed immunofluorescent staining for *Ndr2* on skin before and 3 days after wounding in wild-type (WT) and diabetic mice and included this additional data in the revised version of our manuscript (**Extended data Fig. 7c**). We did not find significant differences in *Ndr2* expression in skin before and after wounding neither in WT nor in diabetic mice. This indicates that wounding does not significantly affect *Ndr2* expression in the skin, and that ectopic *Ndr2* expression in the wound bed likely does not have an impact on wound closure rate. Our data indicate that knocking out *Ndr2* leads to regenerative transcriptomic profiles in DCs and that treatment of wounds with these cells significantly accelerates wound closure. Therefore, we believe that the observed beneficial effects on wound healing are related to our edited DC therapy and the result of our knockout strategy. Our findings demonstrate that *Ndr2* plays an important role in DC biology and maturation which allowed us to reprogram these cells to create a novel cell-based therapy to promote healing of chronic wounds.

10. In Figure 4, wounds treated with control DCs did not heal faster compared to those treated with unseeded hydrogels, which indicates that DCs alone may not provide an additional benefit over collagen-pullulan hydrogels for wound healing. Please discuss the reason in the Discussion section.

- Thank you very much for this valuable comment. Indeed, we did not find significant differences in wound healing after treatment with non-edited DCs compared to blank hydrogels. This finding highlights the strong effect of our gene editing on the DC phenotype. By knocking out *Ndr2* we were able to reprogram these cells and induce regenerative gene expression profiles. Unlike control DCs, these edited DCs significantly accelerated wound closure times when used as a cell-based therapy on wounds in wild-type and diabetic mice. We have added this information to the discussion section of our manuscript.

11. As a powerful and easy-to-use gene editing tool, CRISPR/Cas9 has great potential in scientific research fields. However, this gene editing technology is not perfect, and the off-target effects are a key factor limiting its clinical applications. The authors developed a novel approach for KO of *Ndr2* in DCs, which minimizing off-target effects. However, there is no evidence that this novel approach is completely free of off-target effects. Please mention this in the Discussion section and discuss the other methods to reduce the off-target effects in the Discussion section with latest reference (e.g., <https://doi.org/10.1038/s41586-022-04470-1>).

- We completely agree with the reviewer that Cas9 is not a perfect system. However, CRISPR/Cas9 gene editing has become one of the most effective tools for in vivo research and gene therapy.^{1,34} For example, Intellia and Regeneron have been using Cas9 mediated gene editing, to treat transthyretin (ATTR) amyloidosis, for which they saw a strong therapeutic efficacy with minimal off-target effects^{34,35} Efforts have been made to optimize the specificity of CRISPR/Cas gene editing through protein engineering of Cas9 to increase its specificity or adoption of other Cas molecules (e.g. Cas12a).^{36,37} Future studies will have to test the efficacy of these novel Cas molecules for KO in DCs for the use of cell-based therapies. We have addressed this in the discussion section of our manuscript.

12. Whether the system is safe for skin repair and its safety needs to be tested.

- We completely agree with the reviewer that our study has several limitations that have to be addressed in future studies before clinical translation can be achieved. Rodent models as employed in our study do not exactly recapitulate human wound healing. Therefore, our findings need to be validated in large animal models, such porcine wound models, which more closely mimic the biomechanics and physiology of human skin.^{38,39} Future studies should also test different dosages of cells seeded onto the hydrogels to maximize the therapeutic potential of our technology for wound healing.

Reviewer #4 (Remarks to the Author):

The manuscript shows that deletion of *Ndr2* leads to improved wound healing by BM derived DC populations. The early part of the manuscript is confusing and makes it uncertain which cell populations are actually being affected to produce the effects on wound healing seen in the latter parts of the manuscript. The manuscript employs an impressive array of technologies and robust wound healing modeling. The effect on wounds healing is convincing.

In Figure 1, the authors show 9 clusters but then only describe 3 of them. This is confusing. Are they really distinct cell populations? How do they compare to previously described DC subpopulations? The protocol for preparing the DCs does not provide a purification step or step confirming the DC identity of the cells. The scRNA-seq provides that opportunity. The cells in clusters C4, C6, C7 and C9 (Figure 1) need to be identified. A supplemental Table showing the top differentially expressed genes/cluster should be provided.

- Thank you very much for this valuable comment. To isolate bone marrow-derived dendritic cells (BM-DCs), we followed the protocol by Lutz et al., which is one of the most widely used protocols in the literature and has been shown to enable isolation of BM-DCs with 90-95% purity without requiring further purification steps.⁴⁰⁻⁴² As the reviewer points out, the utility of scRNA-seq in our study provides the opportunity to validate the identity of the cells on a transcriptional level. We apologize for not having included a complete description of all clusters of the UMAP embedding of the BM-DCs in Figure 1. We have added this now to the revised version of our manuscript and also provide a supplemental table with differentially expressed marker genes for the clusters (**Table S1**). Consistent with the findings in the original publication by Lutz et al.,⁴¹ cells in Figure 1 show a DC identity demonstrated by an expression of the characteristic DC markers *Itgax* (encoding CD11c) and *H2-Ab1* (murine MHC-II gene) (**Extended data Fig. 1a**). In line with previous observations on the protein level in BM-DC cultures,⁴¹ cells in our dataset contained both pre-mature DCs (3 clusters) that were characterized by high expression of *Cd34*, *Csf1r* (encoding CD115), *Ccr2*, and *Cx3cr1*.⁴³ In addition, our data included a mature cluster with high expression of *Flt3* (encoding CD135)⁴⁴ and *Ccr7* (**Extended data Fig. 1c**).^{40,45} As the smaller clusters showed some transcriptional overlap, we have adjusted the sensitivity of the clustering according to the suggestion by the reviewer in their comment below. Our new embedding consists of 7 transcriptionally distinct clusters: In addition to the above mentioned pre-mature and mature clusters, other clusters included the *Ndr2*-expressing progenitor cells (now cluster 6); a proliferating cell cluster with high expression of the proliferation marker *Stmn1* (stathmin-1, cluster 2)⁴⁶; cluster 5 characterized by expression of *Tcirg1* which inhibits T cell activation.⁴⁷ The smallest cluster, cluster 7, was a mixed cell population with expression of *Prss34* and *Mcpt8* likely representing

basophilic granulocytes that are often found as a by-product in bone marrow cultures and that are irrelevant for the study (**Extended data Fig. 1d**).⁴⁸

The heterogeneity of BM-DCs generated in vitro under the influence of GM-CSF has been previously described. Several studies have demonstrated that while murine BM-DCs show considerable transcriptional overlap with human monocyte-derived DCs (MoDCs), they still do not recapitulate the distinct subsets of classical DCs (cDCs) that are found under in vivo conditions in peripheral murine tissues.^{40,42} Hence these cells provide less insight for the study of DC biology in vivo, but as demonstrated by our data and previous research are a valuable vehicle for translational cell-based therapies due to their inducible regenerative abilities.

The data provided do not convincingly show that cluster 8 contains the progenitor DC population. As a start the authors do not show that CD34 expression is limited to this population. The rationale for focusing on cluster 8 for the tolerogenic cells is unclear, other than they express higher levels of *Ndr2*.

- We apologize that our manuscript did not clearly demonstrate why cluster 8 (now cluster 6 in the revised manuscript) was identified as the myeloid progenitor population. Apart from a high expression of *Ndr2* and *Cd34*, cluster 8 co-expressed *Csf1r* (encoding CD115), as well as *Flt3* (encoding CD135) and *Clec9a* (encoding DNGR-1). This expression profile is consistent with previously defined myeloid progenitor populations, namely MDP (macrophage and dendritic cell progenitor) and CDP (common dendritic cell progenitor), for which phenotypic overlap has been described in previous studies.⁴⁹⁻⁵¹ To better illustrate this expression profile, we have combined the expression of this gene signature (*Ndr2*, *Csf1r*, *Flt3*, *Clec9a*) using the *AddModuleScore* function in Seurat and projected it onto the UMAP embedding as a FeaturePlot in **Extended Data Figure 1e**. This demonstrates that cluster 6 shows the highest expression of this gene signature and therefore was identified as the progenitor population. In addition, this analysis further confirms that VD3 reduces the expression of this gene signature.

In Fig 1 the authors do not show clustering by treatment except for cluster 8, making it impossible to understand the VD3 changes undergone in other clusters. Figure 3c shows indeed that VD3 cells are found in other clusters and appear to be present in higher proportion in 4 and 7 in addition to cluster 9 (=8 in Fig 1 UMAP). It would be helpful to show this earlier as the reader needs to understand this information associated with Figure 1.

- Thank you very much for this comment. We apologize if this part of our manuscript has not been clear. We show clustering of the entire dataset colored by treatment (untreated vs. VD3 treated BM-DCs) in **Figure 1a**. As suggested by the reviewer, we have added a cell density plot for the dataset of Figure 1 to the revised version of the manuscript and show it now in **Extended Data Figure 1b**.

For both the UMAP clustering in Figures 1 and 3, my tendency would be for the UMAP clusters to be either: 1) combined into the subsets the investigators are able to identify clearly, 2) adjust the sensitivity of the UMAP clustering to more accurately reflect the known subsets or 3) provide feature plots or violins showing the distinctive gene expression associated with the different clusters and mention that the implications of the different DC clusters is unknown.

- Thank you very much for this great comment. As the reviewer suggested, we have included a more detailed characterization of the clusters of the UMAP embedding of Figures 1 and 3 in the revised version of our manuscript. With regard to the clusters of Figure 1, we refer to our response to the reviewer's question above. As for the data in

Figure 1b, distinct marker expression allowed us to differentiate the DC clusters in the dataset shown in Figure 3b into pre-mature and mature subsets. We have included feature plots showing the expression of specific markers in **Extended Data Figure 5** of our revised manuscript and have provided a supplemental table for the DEG of the clusters (**Table S4**). Most *Ndr2*-KO cells were found in cluster 1 which was identified as a premature DC cluster with strong expression of *Csf1r*, *Ccr2*, and *Cx3cr1* (**Extended Data Fig. 5a**) as well as regenerative markers *Fn1*, *Arg1*, *Mmp12* among its top DEGs (**Extended Data Fig. 5b**). By contrast, most control and VD3 treated cells localized to cluster 0 which was characterized a mature cluster with strong expression of maturation markers such as *Ccr7*, *Flt3*, *Cd80*, and *Cd83* (**Extended Data Fig. 5c**). Cluster 3 was identified as a smaller mature cluster which mostly contained *Ndr2*-KO cells and strongly expressed *Nr4a3* (**Extended Data Fig. 5d**). Clusters 2, 7, and 8 were identified as pre-mature clusters that mostly contained control and VD3 treated cells and expressed *Cx3cr1* and *Csf1r* (cluster 7) as well as *S100a9* (clusters 2 and 8) and *Stmn1* (cluster 8) (**Extended Data Fig. 5a, e**). Cluster 4 contained the *Stmn1* expressing proliferating cells mostly from the VD3 treated group, that had been identified in Figure 1b (**Extended Data Fig. 5d**). Cluster 11 represented a very small cluster of mixed cells as a by-product of the bone marrow cultures. We have added this information to the revised manuscript. Previous studies have found that BM-DCs that are generated under GM-CSF in vitro are a model system and do not accurately represent classical DC subsets found under in vivo conditions.^{40,42} As the reviewer correctly points out, the exact implications of the different DC clusters in our dataset remain incompletely understood and will require further investigation.

Controls for the experiments in Figure 1i are lacking. What is the effect of LPS and VD3 directly on the endothelial cell cultures, or cultures with no DC and no treatment. The ECs shown at baseline appear very poorly formed.

- Thank you very much for this comment. We apologize that our manuscript may have been unclear with regard to the methodology and different groups of the cell culture experiment in Figure 1g-i. The purpose of this experiment was to investigate the impact of dendritic cells (DCs) in which *Ndr2* was pharmacologically inhibited on endothelial cell (EC) tube formation. Therefore, the DC monocultures were pre-treated with drugs that are known to inhibit *Ndr2* (lipopolysaccharide and vitamin D3 as well as dexamethasone) for 24 hours. After pre-treatment, the cells were washed and then co-cultured with ECs. As such, the drugs were not directly added to the co-culture and any effect observed on EC tube formation in the co-cultures is therefore related to the impact of the pre-treated DCs. Hence, we believe that investigating the effect of LPS and VD3 directly on EC cultures is certainly interesting, however, would not provide additional insight on the role of *Ndr2* on the pro-angiogenic capacity of DCs. The EC monocultures that are shown in **Extended data figure 2c (right panel)** serve as a negative control. These ECs were grown in basal media only, which is depleted of any growth factors that are required for tube formation, hence these cells are not expected to form tubes, which is consistent with our observation and previously published protocols.⁵²

Please label the axes of Figure 2c.

- We apologize for having missed the axis labels in **Figure 2c** and have added the labels.

Altered gene expression by the fibroblasts should be made more granular. In the absence of keys in Fig 5e and 5h, I find these panels impossible to interpret. As a reader, I am anxious to know more broadly the genes upregulated in the Ndr2 k/o DC-treated wounds. This could be provided by a table, and/or by a more complete labeling of the genes in 5e. In any event supplemental tables providing the DEGs for each cluster should be provided.

- Thank you very much for this helpful comment. As suggested, we have added a volcano plot with gene labels in **Extended data Fig. 8a**. We have also provided supplemental **Tables S5 – S8** including the differentially expressed marker genes for both groups and for each cell type.

References

1. Lu Y, Xue J, Deng T, et al. Safety and feasibility of CRISPR-edited T cells in patients with refractory non-small-cell lung cancer. *Nat Med*. May 2020;26(5):732-740. doi:10.1038/s41591-020-0840-5
2. Atkins A, Chung CH, Allen AG, et al. Off-Target Analysis in Gene Editing and Applications for Clinical Translation of CRISPR/Cas9 in HIV-1 Therapy. *Front Genome Ed*. 2021;3:673022. doi:10.3389/fgeed.2021.673022
3. Bentley DR, Balasubramanian S, Swerdlow HP, et al. Accurate whole human genome sequencing using reversible terminator chemistry. *Nature*. Nov 6 2008;456(7218):53-9. doi:10.1038/nature07517
4. Wang S, Ren S, Bai R, et al. No off-target mutations in functional genome regions of a CRISPR/Cas9-generated monkey model of muscular dystrophy. *J Biol Chem*. Jul 27 2018;293(30):11654-11658. doi:10.1074/jbc.AC118.004404
5. Stadtmauer EA, Fraietta JA, Davis MM, et al. CRISPR-engineered T cells in patients with refractory cancer. *Science*. Feb 28 2020;367(6481)doi:10.1126/science.aba7365
6. Behr M, Zhou J, Xu B, Zhang H. In vivo delivery of CRISPR-Cas9 therapeutics: Progress and challenges. *Acta Pharm Sin B*. Aug 2021;11(8):2150-2171. doi:10.1016/j.apsb.2021.05.020
7. Singh K, Cornell CS, Jackson R, et al. CRISPR/Cas9 generated knockout mice lacking phenylalanine hydroxylase protein as a novel preclinical model for human phenylketonuria. *Sci Rep*. Mar 31 2021;11(1):7254. doi:10.1038/s41598-021-86663-8
8. Nguyen TH, Anegon I. Successful correction of hemophilia by CRISPR/Cas9 genome editing in vivo: delivery vector and immune responses are the key to success. *EMBO Mol Med*. May 2016;8(5):439-41. doi:10.15252/emmm.201606325
9. Deng S, Li X, Liu S, et al. Codelivery of CRISPR-Cas9 and chlorin e6 for spatially controlled tumor-specific gene editing with synergistic drug effects. *Sci Adv*. Jul 2020;6(29):eabb4005. doi:10.1126/sciadv.abb4005
10. Gee P, Lung MSY, Okuzaki Y, et al. Extracellular nanovesicles for packaging of CRISPR-Cas9 protein and sgRNA to induce therapeutic exon skipping. *Nat Commun*. Mar 13 2020;11(1):1334. doi:10.1038/s41467-020-14957-y
11. Chou SJ, Yang P, Ban Q, et al. Dual Supramolecular Nanoparticle Vectors Enable CRISPR/Cas9-Mediated Knockin of Retinoschisin 1 Gene-A Potential Nonviral Therapeutic Solution for X-Linked Juvenile Retinoschisis. *Adv Sci (Weinh)*. May 2020;7(10):1903432. doi:10.1002/advs.201903432
12. Sen CK, Gordillo GM, Roy S, et al. Human skin wounds: a major and snowballing threat to public health and the economy. *Wound Repair Regen*. Nov-Dec 2009;17(6):763-71. doi:10.1111/j.1524-475X.2009.00543.x
13. Nussbaum SR, Carter MJ, Fife CE, et al. An Economic Evaluation of the Impact, Cost, and Medicare Policy Implications of Chronic Nonhealing Wounds. *Value Health*. Jan 2018;21(1):27-32. doi:10.1016/j.jval.2017.07.007
14. desJardins-Park HE, Gurtner GC, Wan DC, Longaker MT. From Chronic Wounds to Scarring: The Growing Healthcare Burden of Under- and Over-healing Wounds. *Adv Wound Care (New Rochelle)*. Sep 15 2021;doi:10.1089/wound.2021.0039

15. Sivaraj D, Chen K, Chattopadhyay A, et al. Hydrogel Scaffolds to Deliver Cell Therapies for Wound Healing. *Front Bioeng Biotechnol.* 2021;9:660145. doi:10.3389/fbioe.2021.660145
16. Levine BL, Miskin J, Wonnacott K, Keir C. Global Manufacturing of CAR T Cell Therapy. *Mol Ther Methods Clin Dev.* Mar 17 2017;4:92-101. doi:10.1016/j.omtm.2016.12.006
17. Health NCIatNlo. CAR T Cells: Engineering Patients' Immune Cells to Treat Their Cancers. <https://www.cancer.gov/about-cancer/treatment/research/car-t-cells>.
18. Gurtner GC, Werner S, Barrandon Y, Longaker MT. Wound repair and regeneration. *Nature.* May 15 2008;453(7193):314-21. doi:10.1038/nature07039
19. Henn D, Chen K, Fehlmann T, et al. Xenogeneic skin transplantation promotes angiogenesis and tissue regeneration through activated Trem2(+) macrophages. *Sci Adv.* Dec 3 2021;7(49):eabi4528. doi:10.1126/sciadv.abi4528
20. Huber-Lang M, Lambris JD, Ward PA. Innate immune responses to trauma. *Nat Immunol.* Apr 2018;19(4):327-341. doi:10.1038/s41590-018-0064-8
21. Kim SY, Nair MG. Macrophages in wound healing: activation and plasticity. *Immunol Cell Biol.* Mar 2019;97(3):258-267. doi:10.1111/imcb.12236
22. Han Z, Chen Y, Zhang Y, et al. MiR-21/PTEN Axis Promotes Skin Wound Healing by Dendritic Cells Enhancement. *J Cell Biochem.* Oct 2017;118(10):3511-3519. doi:10.1002/jcb.26026
23. Maschalidi S, Mehrotra P, Keceli BN, et al. Targeting SLC7A11 improves efferocytosis by dendritic cells and wound healing in diabetes. *Nature.* Jun 2022;606(7915):776-784. doi:10.1038/s41586-022-04754-6
24. Vinish M, Cui W, Stafford E, et al. Dendritic cells modulate burn wound healing by enhancing early proliferation. *Wound Repair Regen.* Jan-Feb 2016;24(1):6-13. doi:10.1111/wrr.12388
25. Thurner B, Roder C, Dieckmann D, et al. Generation of large numbers of fully mature and stable dendritic cells from leukapheresis products for clinical application. *J Immunol Methods.* Feb 1 1999;223(1):1-15. doi:10.1016/s0022-1759(98)00208-7
26. Choi SC, Kim KD, Kim JT, et al. Expression and regulation of NDRG2 (N-myc downstream regulated gene 2) during the differentiation of dendritic cells. *FEBS Lett.* Oct 23 2003;553(3):413-8. doi:10.1016/s0014-5793(03)01030-5
27. Riboldi E, Musso T, Moroni E, et al. Cutting edge: proangiogenic properties of alternatively activated dendritic cells. *J Immunol.* Sep 1 2005;175(5):2788-92. doi:10.4049/jimmunol.175.5.2788
28. Henn D, Abu-Halima M, Wermke D, et al. MicroRNA-regulated pathways of flow-stimulated angiogenesis and vascular remodeling in vivo. *J Transl Med.* Jan 11 2019;17(1):22. doi:10.1186/s12967-019-1767-9
29. Liu S, Yang P, Kang H, et al. NDRG2 induced by oxidized LDL in macrophages antagonizes growth factor productions via selectively inhibiting ERK activation. *Biochim Biophys Acta.* Feb 2010;1801(2):106-13. doi:10.1016/j.bbali.2009.09.022
30. Hu W, Fan C, Jiang P, et al. Emerging role of N-myc downstream-regulated gene 2 (NDRG2) in cancer. *Oncotarget.* Jan 5 2016;7(1):209-23. doi:10.18632/oncotarget.6228
31. Liu Z, Wu H, Huang S. Role of NGF and its receptors in wound healing (Review). *Exp Ther Med.* Jun 2021;21(6):599. doi:10.3892/etm.2021.10031

32. Kim MH, Wu WH, Choi JH, et al. Galectin-1 from conditioned medium of three-dimensional culture of adipose-derived stem cells accelerates migration and proliferation of human keratinocytes and fibroblasts. *Wound Repair Regen*. Dec 2018;26 Suppl 1:S9-S18. doi:10.1111/wrr.12579
33. Lin YT, Chen JS, Wu MH, et al. Galectin-1 accelerates wound healing by regulating the neuropilin-1/Smad3/NOX4 pathway and ROS production in myofibroblasts. *J Invest Dermatol*. Jan 2015;135(1):258-268. doi:10.1038/jid.2014.288
34. Gillmore JD, Gane E, Taubel J, et al. CRISPR-Cas9 In Vivo Gene Editing for Transthyretin Amyloidosis. *N Engl J Med*. Aug 5 2021;385(6):493-502. doi:10.1056/NEJMoa2107454
35. Carvalho T. CRISPR-Cas9 hits its target in amyloidosis. *Nat Med*. Dec 2022;28(12):2438. doi:10.1038/d41591-022-00101-4
36. Oakes BL, Nadler DC, Savage DF. Protein engineering of Cas9 for enhanced function. *Methods Enzymol*. 2014;546:491-511. doi:10.1016/B978-0-12-801185-0.00024-6
37. Zhou J, Chen P, Wang H, et al. Cas12a variants designed for lower genome-wide off-target effect through stringent PAM recognition. *Mol Ther*. Jan 5 2022;30(1):244-255. doi:10.1016/j.ymthe.2021.10.010
38. Chen K, Henn D, Januszyk M, et al. Disrupting mechanotransduction decreases fibrosis and contracture in split-thickness skin grafting. *Sci Transl Med*. May 18 2022;14(645):eabj9152. doi:10.1126/scitranslmed.abj9152
39. Chen K, Kwon SH, Henn D, et al. Disrupting biological sensors of force promotes tissue regeneration in large organisms. *Nat Commun*. Sep 6 2021;12(1):5256. doi:10.1038/s41467-021-25410-z
40. Lutz MB, Inaba K, Schuler G, Romani N. Still Alive and Kicking: In-Vitro-Generated GM-CSF Dendritic Cells! *Immunity*. Jan 19 2016;44(1):1-2. doi:10.1016/j.immuni.2015.12.013
41. Lutz MB, Kukutsch N, Ogilvie AL, et al. An advanced culture method for generating large quantities of highly pure dendritic cells from mouse bone marrow. *J Immunol Methods*. Feb 1 1999;223(1):77-92. doi:10.1016/s0022-1759(98)00204-x
42. Lutz MB, Strobl H, Schuler G, Romani N. GM-CSF Monocyte-Derived Cells and Langerhans Cells As Part of the Dendritic Cell Family. *Front Immunol*. 2017;8:1388. doi:10.3389/fimmu.2017.01388
43. Nakano H, Lyons-Cohen MR, Whitehead GS, Nakano K, Cook DN. Distinct functions of CXCR4, CCR2, and CX3CR1 direct dendritic cell precursors from the bone marrow to the lung. *J Leukoc Biol*. May 2017;101(5):1143-1153. doi:10.1189/jlb.1A0616-285R
44. Harada S, Kimura T, Fujiki H, et al. Flt3 ligand promotes myeloid dendritic cell differentiation of human hematopoietic progenitor cells: possible application for cancer immunotherapy. *Int J Oncol*. Jun 2007;30(6):1461-8.
45. Dieu MC, Vanbervliet B, Vicari A, et al. Selective recruitment of immature and mature dendritic cells by distinct chemokines expressed in different anatomic sites. *J Exp Med*. Jul 20 1998;188(2):373-86. doi:10.1084/jem.188.2.373
46. Brown CC, Gudjonson H, Pritykin Y, et al. Transcriptional Basis of Mouse and Human Dendritic Cell Heterogeneity. *Cell*. Oct 31 2019;179(4):846-863 e24. doi:10.1016/j.cell.2019.09.035
47. Makaryan V, Rosenthal EA, Bolyard AA, et al. TCIRG1-associated congenital neutropenia. *Hum Mutat*. Jul 2014;35(7):824-7. doi:10.1002/humu.22563

48. Ugajin T, Kojima T, Mukai K, et al. Basophils preferentially express mouse Mast Cell Protease 11 among the mast cell tryptase family in contrast to mast cells. *J Leukoc Biol.* Dec 2009;86(6):1417-25. doi:10.1189/jlb.0609400
49. Waskow C, Liu K, Darrasse-Jeze G, et al. The receptor tyrosine kinase Flt3 is required for dendritic cell development in peripheral lymphoid tissues. *Nat Immunol.* Jun 2008;9(6):676-83. doi:10.1038/ni.1615
50. Poltorak MP, Schraml BU. Fate mapping of dendritic cells. *Front Immunol.* 2015;6:199. doi:10.3389/fimmu.2015.00199
51. Cabeza-Cabrerizo M, Cardoso A, Minutti CM, Pereira da Costa M, Reis ESC. Dendritic Cells Revisited. *Annu Rev Immunol.* Apr 26 2021;39:131-166. doi:10.1146/annurev-immunol-061020-053707
52. DeCicco-Skinner KL, Henry GH, Cataisson C, et al. Endothelial cell tube formation assay for the in vitro study of angiogenesis. *J Vis Exp.* Sep 1 2014;(91):e51312. doi:10.3791/51312

REVIEWERS' COMMENTS

Reviewer #2 (Remarks to the Author):

The authors have significantly improved the revised manuscript. However, there are a few more questions to address here.

Could Cas9-Mediated Knockout of Ndr2 potentially cause abnormal activity in other cells and sites? I am still not entirely convinced if the Ndr2 gene could be only a downstream phenomenon rather than a significant factor in promoting the vascularization of ECs after VD3 treatment of DCs?

Reviewer #3 (Remarks to the Author):

No more comments.

Reviewer #4 (Remarks to the Author):

The "e" label for panel 5e is missing. My previous recommendations and concerns have been addressed adequately.

Reviewer #5 (Remarks to the Author):

The questions of referee 1 have been addressed and answered.

There is a figure duplication in Figure 4 h Day 3.

Immunofluorescence staining of untreated and CRISPR-edited cells in Figure 2 e needs a Ndr2-independent DC marker.

Image quality of Figure 1 h (VD3/LPS-DC + EC) is rather mediocre for a confocal image; furthermore the number of both DC and HUVEC cells seems dramatically different in both conditions, whereas their numbers should be identical but only tube formation should be different.

Reviewer #2 Comments:

The authors have significantly improved the revised manuscript. However, there are a few more questions to address here. Could Cas9-Mediated Knockout of *Ndr2* potentially cause abnormal activity in other cells and sites?

- We thank the reviewer for this comment. We have used Cas9-RNP technology to knock out *Ndr2* in bone marrow-derived DCs in vitro. These engineered DCs were then seeded onto hydrogels for delivery of the cells onto wounds where they exert their regenerative effect to promote wound healing. Hence, the Cas9-RNP knockout was only applied to the cells in vitro before bringing them into the wound via hydrogels. In vivo knockout technologies were not used in the study. Therefore, we can exclude any additional effect of the CRISPR-KO technology on other cells than the DCs in vitro.

I am still not entirely convinced if the *Ndr2* gene could be only a downstream phenomenon rather than a significant factor in promoting the vascularization of ECs after VD3 treatment of DCs?

- Thank you for this comment. We completely agree with the reviewer that VD3 treatment transcriptionally affects multiple genes in DCs, some of which may have a beneficial effect on angiogenesis while others may just be a downstream phenomenon. The role of *Ndr2* as a potent inhibitor of growth factor expression, angiogenesis, and cell proliferation has been established in previous studies.¹⁻⁵ Therefore, *Ndr2* appeared to be an interesting target for further investigation. Single-cell RNA-seq of VD3 treated and control DCs revealed that *Ndr2* marks a specific progenitor population of MDP (macrophage and dendritic cell progenitor) and CDP (common dendritic cell progenitor). Using our Cas9 RNP knockout strategy, we demonstrated that knockout of *Ndr2* shifts DC populations to a more premature state and induced regenerative and pro-angiogenic transcriptomic profiles in these cells. Using in vivo experiments on murine excisional wounds, we show that these edited DCs promote angiogenesis of the wound bed. In light of this data, we believe that KO of

Ndr2 in DCs is a promising strategy for cell-based therapies to promote angiogenesis and healing of diabetic and non-diabetic wounds.

Reviewer #3 Comments:

No more comments.

- Thank you to the reviewer for reading our revised manuscript.

Reviewer #4 Comments:

The "e" label for panel 5e is missing. My previous recommendations and concerns have been addressed adequately.

- Thank you for bringing this to our attention. We have added the label to Figure panel 5e.

Reviewer #5 Comments:

The questions of referee 1 have been addressed and answered.

There is a figure duplication in Figure 4 h Day 3.

- Thank you very much for bringing this to our attention. We apologize for this error and have added the correct image.

Immunofluorescence staining of untreated and CRISPR-edited cells in Figure 2 e needs a Ndr2-independent DC marker.

- Thank you very much for this comment. This question asks for confirmation of cell identity after CRISPR knockout. We believe single-cell RNA sequencing is more suitable to address this question, since it allows for a more comprehensive analysis of different markers compared to immunofluorescent staining of a single marker. Single-cell RNA-seq of Ndr2-knockout and control DCs showed that the edited cells retained their DC identity and express common DC markers such as *Ptprc* (encoding CD45), *Itgax* (encoding CD11c), and *H2-Ab1* (encoding MHCII). We have added this information to the revised manuscript and to Extended Data Figure 3b.

Image quality of Figure 1 h (VD3/LPS-DC + EC) is rather mediocre for a confocal image; furthermore the number of both DC and HUVEC cells seems dramatically different in both conditions, whereas their numbers should be identical but only tube formation should be different.

- Thank you very much for this comment. We have replaced the image with a higher quality confocal image in the new Figure 1h. Cell counts for this experiment were standardized and HUVECs were plated at a concentration of 3.5×10^4 in 200 μ l media per well. DCs were added to the cultures in a 1:1 ratio.

References

1. Choi SC, Kim KD, Kim JT, et al. Expression and regulation of NDRG2 (N-myc downstream regulated gene 2) during the differentiation of dendritic cells. *FEBS Lett.* Oct 23 2003;553(3):413-8. doi:10.1016/s0014-5793(03)01030-5
 2. Riboldi E, Musso T, Moroni E, et al. Cutting edge: proangiogenic properties of alternatively activated dendritic cells. *J Immunol.* Sep 1 2005;175(5):2788-92. doi:10.4049/jimmunol.175.5.2788
 3. Henn D, Abu-Halima M, Wermke D, et al. MicroRNA-regulated pathways of flow-stimulated angiogenesis and vascular remodeling in vivo. *J Transl Med.* Jan 11 2019;17(1):22. doi:10.1186/s12967-019-1767-9
 4. Liu S, Yang P, Kang H, et al. NDRG2 induced by oxidized LDL in macrophages antagonizes growth factor productions via selectively inhibiting ERK activation. *Biochim Biophys Acta.* Feb 2010;1801(2):106-13. doi:10.1016/j.bbaliip.2009.09.022
 5. Hu W, Fan C, Jiang P, et al. Emerging role of N-myc downstream-regulated gene 2 (NDRG2) in cancer. *Oncotarget.* Jan 5 2016;7(1):209-23. doi:10.18632/oncotarget.6228
-